# Strong metal-support interactions induced by an ultrafast laser

Jian Zhang[1,3], Dezhi Zhu[2,3], Jianfeng Yan [2✉] & Chang-An Wang [1✉]

Supported metal catalysts play a crucial role in the modern industry. Constructing strong metal-support interactions (SMSI) is an effective means of regulating the interfacial properties of noble metal-based supported catalysts. Here, we propose a new strategy of ultrafast laser-induced SMSI that can be constructed on a $CeO_2$-supported Pt system by confining electric field in localized interface. The nanoconfined field essentially boosts the formation of surface defects and metastable $CeO_x$ migration. The SMSI is evidenced by covering Pt nanoparticles with the $CeO_x$ thin overlayer and suppression of CO adsorption. The overlayer is permeable to the reactant molecules. Owing to the SMSI, the resulting $Pt/CeO_2$ catalyst exhibits enhanced activity and stability for CO oxidation. This strategy of constructing SMSI can be extended not only to other noble metal systems (such as $Au/TiO_2$, $Pd/TiO_2$, and $Pt/TiO_2$) but also on non-reducible oxide supports (such as $Pt/Al_2O_3$, $Au/MgO$, and $Pt/SiO_2$), providing a universal way to engineer and develop high-performance supported noble metal catalysts.

[1] State Key Laboratory of New Ceramics and Fine Processing, School of Materials Science and Engineering, Tsinghua University, Beijing 100084, China. [2] State Key Laboratory of Tribology, Department of Mechanical Engineering, Tsinghua University, Beijing 100084, China. [3] These authors contributed equally: Jian Zhang, Dezhi Zhu. ✉email: yanjianfeng@tsinghua.edu.cn; wangca@tsinghua.edu.cn

Supported noble metal catalysts, dispersed the noble metal nanoparticles (NPs) on an oxide support with high specific surface area, are one of the most important heterogeneous catalysts[1,2]. They have a wide range of applications in the fields of energy conversion, chemical production, and exhaust gas purification, and play a vital role in the global economy[3]. In the original catalytic studies, oxide support was considered chemically inert, serving only to anchor and disperse the active component. In the late 1970s, Tauster et al. found that the adsorption of small molecules (such as CO, $H_2$) was significantly suppressed by the high temperature reduction of platinum group metals (PGMs) supported on $TiO_2$, and that the cause of this phenomenon was not due to sintering or poisoning of the noble metals. Therefore, this unusual interaction between metal and reducible metal oxides was named "strong metal-support interaction" (SMSI)[4,5]. The formation of SMSI effect can profoundly affect the electronic structure and geometry of catalysts, thus altering their activity, selectivity, and stability, and has therefore been extensively investigated in recent decades[6–8]. It is generally accepted that the classical SMSI effect is associated with the encapsulation of the metal by the support species. The overcoating consists of several atomic layers, usually in an amorphous state, and has a dynamic structure in different gas atmospheres[9,10]. In general, SMSI effect tends to occur between reducible metal oxides with relatively low surface energy (such as $TiO_2$, $V_2O_5$, $Nb_2O_5$, and $Ta_2O_5$) and Pt group metals, where the thermodynamic driving force is the minimization of surface energy[7,11,12]. The most classical method for constructing SMSI effects is high-temperature hydrogen treatment. The lattice oxygen on the oxide support surface is abstracted to form a suboxide and migrates to the metal surface to form a stable overlayer[4,5,13,14]. Although the study of constructing SMSI by hydrogen reduction has been developed for decades, the insufficiency of this approach is significant, focusing on the following aspects: (i) Catalyst systems are mostly limited to reducible metal oxide loaded Pt group metals. (ii) The formation of SMSI effects may be accompanied by sintering of the metal particles, as the temperature of thermal reduction is usually higher than 500 °C[11]. In order to break the bottleneck, classical SMSI was achieved between Au and non-reducible MgO through $CO_2$-induced activation of the oxide surface[15]. Wang et al. reported a strong interaction between Au and $TiO_2$ induced by melamine. The presence of a permeable $TiO_x$ overlayer ensured that the catalyst maintained high catalytic activity even after calcination at 800 °C[16]. Recently, Xiao and co-workers[17] proposed a wet-chemistry methodology to construct SMSI on titania-supported Au NPs (Au/$TiO_2$-wcSMSI), avoiding the necessity of high-temperature treatment[17]. Thermally induced reactions in specific gaseous atmospheres are generally required in traditional procedures. Moreover, constructing SMSI with the above methodology is usually not widely generalizable. Therefore, new methods that enable the universal construction of SMSI in various catalytic systems under ambient conditions are still essential to design high-performance catalysts and understand SMSI effects in more depth.

Laser ablation in liquid (LAL) is generally accepted as a universal, green, and one-step method for synthesizing metastable functional nanomaterials with novel properties through photo-induced localized physical/chemical processes[18–24]. Recently, LAL has been utilized to fabricate $TiO_2$ NPs for solar energy conversion and environmental remediation[25–27]. Self-doped $TiO_2$ nanocrystals were synthesized using a nanosecond laser, and $Ti^{3+}$/oxygen vacancies were successfully induced in the $TiO_2$ NPs, which resulted in high photocatalytic activity. However, the high temperature and high pressure caused by the nanosecond or longer pulse irradiation may induce unfavorable phase transformation of anatase to rutile $TiO_2$ with inferior properties[28]. For aggregated CuO NPs, the photothermal effects involved in long-pulse laser irradiation may cause sintering with the sacrifice of active sites[29]. For non-aggregated $TiO_2$ NPs, when using nanosecond laser irradiation, a gradual decrease of photocurrent may result from the formation of bulk defects due to thermally initiated isochoric melting. When using a small number of picosecond pulses, the performances improved by a factor of two[30,31]. The unique characteristics of ultrafast laser, ultrahigh intensity and ultrashort pulse duration, can induce nonlinear absorption, which may provide a solution to these challenges. When the high-energy laser interacts with the metal oxides dispersed in the solution, the surface structure is reconfigured. Oxygen vacancies formed on the surface of the metal oxides, leaving the surface species in a sub-oxide state, thus achieving activation of the metal oxides surface. The above phenomenon may provide the preliminary conditions for the successful construction of the SMSI effect.

In this study, we proposed a novel strategy to induce SMSI in $CeO_2$-supported Pt NPs based on ultrafast laser excitation. We succeeded in creating porous overlayers of $CeO_x$ on Pt NPs, which exhibit superior catalytic activity and stability (Fig. 1). The mechanism underlying the SMSI formation was revealed, and depended on the localized energy deposition. To the best of our knowledge, this is the first report on the laser-induced SMSI, and our approach can be facilely extended to other material systems (such as Pt/$TiO_2$, Pd/$TiO_2$, Au/$TiO_2$, Pt/$Al_2O_3$, Au/MgO, and Pt/$SiO_2$).

## Results

**Ultrafast laser induced strong metal-support interactions.** Figure 2a shows the schematic illustration of the ultrafast laser irradiation of Pt/$CeO_2$ NPs, and the nanoconfined electric field between Pt NPs and $CeO_2$ were created. The calculated electric field distributions on multiple and single Pt/$CeO_2$ excited at an

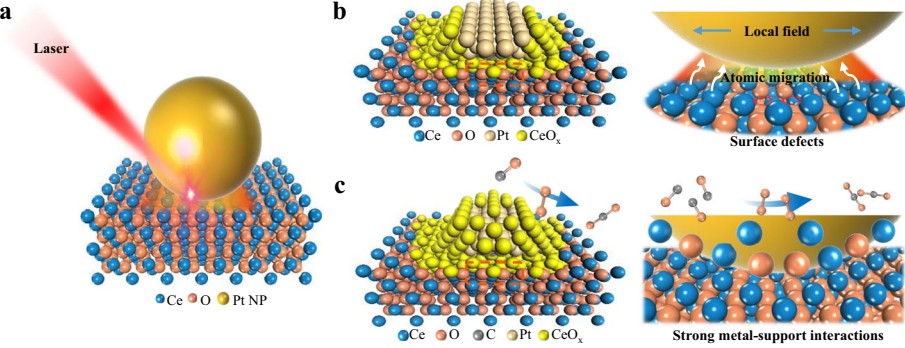

**Fig. 1 Schematic of the ultrafast laser-induced SMSI in Pt/CeO₂. a** Pt/$CeO_2$ nanostructure irradiated at ultrafast laser. **b** The surface defects and metastable $CeO_x$ migration induced by local field. **c** The high catalytic stability obtained from the laser-induced SMSI.

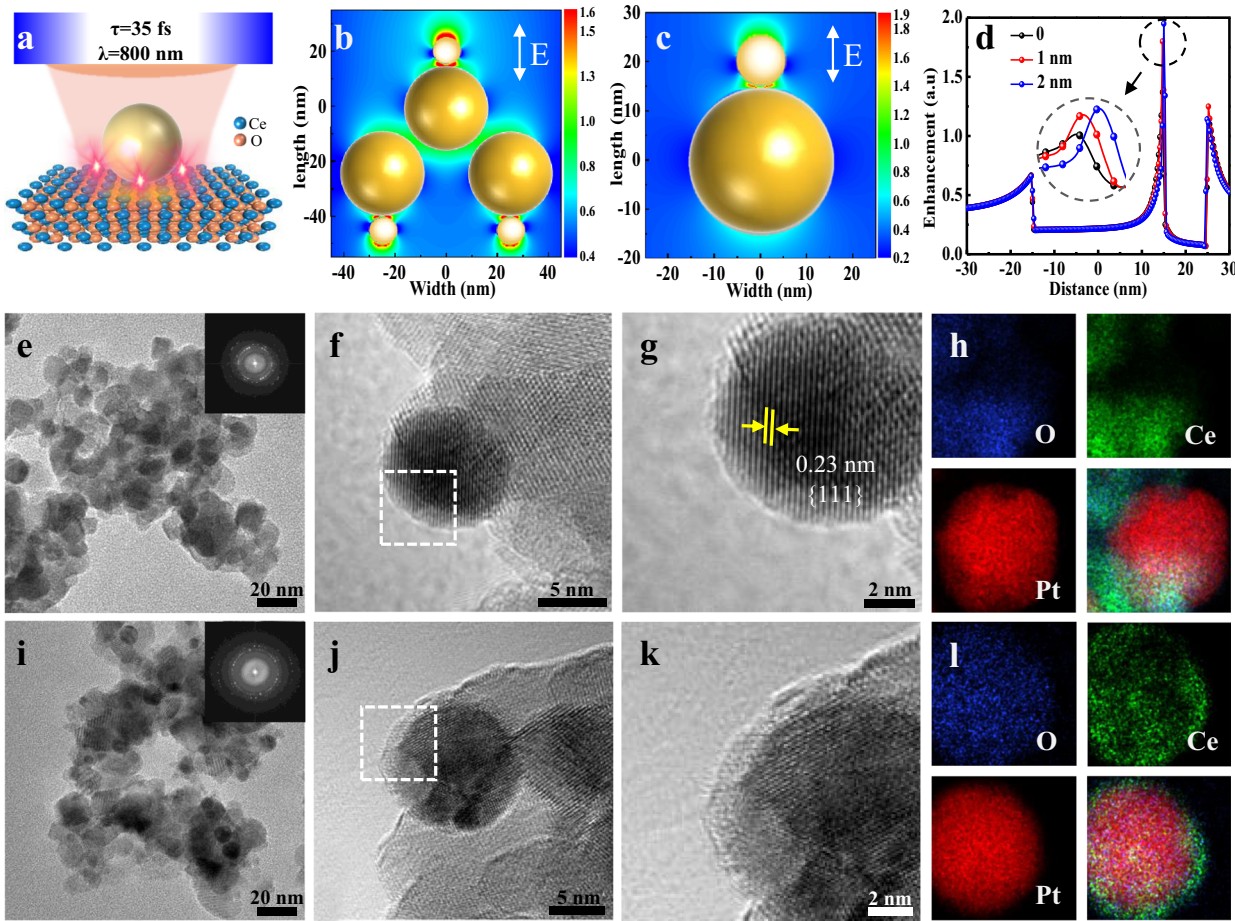

**Fig. 2 Ultrafast laser induced SMSI. a** Schematic of the setup for ultrafast laser irradiation of Pt/CeO₂. **b, c** Calculated electric field distribution and (**d**) the enhancement profiles along the longitudinal axis of the Pt/CeO₂ excited at an 800-nm wavelength with parallel polarization. Inset is the enlarged image of the enhanced electric field at the interface between Pt and CeO₂ (width = 0, 1, 2 nm). **e, f** TEM image of as-prepared Pt/CeO₂. Inset is the FFT. **g** The enlarged image of the areas marked in (**f**. **h**) EDS elemental mapping of as-prepared Pt/CeO₂. **i, j** TEM image of laser-irradiated Pt/CeO₂. Inset is the FFT. **k** The enlarged image of the areas marked in (**j**). **l** EDS elemental mapping of laser-irradiated Pt/CeO₂.

800-nm wavelength with parallel polarization are presented (Fig. 2b, c). Figure 2d shows the calculated electric field enhancement profiles along the longitudinal axis of Pt/CeO₂ excited at parallel polarization with the wavelength of 800 nm. Note that the electric field intensity at the interface between Pt and CeO₂ was significantly larger than that on other regions. It means that the laser energy was mainly deposited on the interface. Pt/CeO₂ catalysts were prepared by the NaBH₄ reduction method. According to the transmission electron microscopy (TEM) images of the Pt/CeO₂ (Fig. 2e, f), the Pt NPs exhibited a spherical shape with a diameter of about 10 nm. As indicated by the high-resolution transmission electron microscopy (HRTEM) images (Fig. 2g), the interface contrast between the Pt and CeO₂ NPs can clearly be observed, and no material was deposited on the surface of Pt NPs. The nanostructure was confirmed by energy-dispersive X-ray spectroscopy (EDS) elemental mapping (Fig. 2h). In the EDS profiles, there was no other element signal on the region where it exhibited an obvious Pt element signal. Upon ultrafast laser irradiation, Pt NPs were encapsulated by a thin overlayer (Fig. 2i–k). Moreover, Pt NPs with different sizes can be covered by thin overlayers (Supplementary Fig. 1). As shown in the EDS elemental mapping (Fig. 2l), cerium and oxygen were detected in the overlayers. It suggests that the surface of the Pt NPs was decorated with material coming from the support after ultrafast laser irradiation. In addition, a small number of NPs showed signs of growth after laser treatment, but

the overall NP size distribution did not change significantly (Supplementary Fig. 2). The above results indicate that the ultrafast laser irradiation successfully induced the overlayer structure formation in CeO₂-supported Pt NPs.

The electron energy loss spectroscopy (EELS) was employed to further study the nature of the overlayer in laser-irradiated Pt/CeO₂ (Fig. 3a, b). Both on the support (region 4) and Pt NP (regions 2 and 3) surface, two obvious peaks located at 882 and 901 eV in Ce spectra, which correspond to $M_5$ and $M_4$ edges, respectively. These two peaks come from electrons transitioning from the spin–orbit splitting energy levels $3d_{5/2}$ and $3d_{3/2}$ to the unoccupied $4f$ state. The previous study showed that the intensity ratios between $M_4$ and $M_5$ edges were 1.12 and 0.75 for $Ce^{4+}$ and $Ce^{3+}$, respectively[32,33]. The increase in $Ce^{3+}$ concentration in Ce-containing oxides with fluorite or modified-fluorite structure is reflected in a decrease in the $M_4/M_5$ intensity ratio. On the support (CeO₂), the $M_4/M_5$ value is about 1.1, while on the Pt NP the value is about 0.95. This result provides substantial evidence that the overlayer is $CeO_x$ ($x < 2$) species, which is well consistent with the previous studies. In addition, EELS spectra were performed on relatively small size Pt NP (about 4 nm) and corresponding interfacial perimeter (Supplementary Fig. 3), cerium species mainly exists in the trivalent state in these regions. It is worth mentioning that the EELS results also indirectly proved that the surface reconstruction of the support after laser irradiation resulted in the formation of metastable

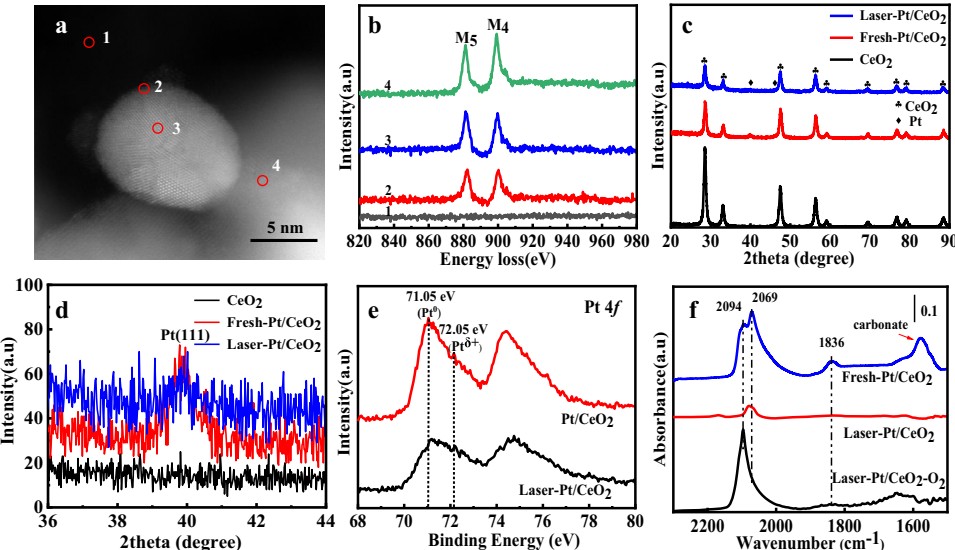

**Fig. 3 Structural characterizations. a** High-angle annular dark-field scanning transmission electron microscopy (HAADF-STEM) image of laser-irradiated Pt/CeO₂. **b** EELS spectrum of laser-irradiated Pt/CeO₂. **c, d** XRD spectra of Pt/CeO₂ and laser-irradiated Pt/CeO₂. **e** XPS spectra of fresh Pt/CeO₂ and laser-irradiated Pt/CeO₂. **f** In situ CO-DRIFT of fresh Pt/CeO₂ and laser-irradiated Pt/CeO₂.

$CeO_x$ species. Figure 3c, d illustrates the XRD spectra of the samples. After laser irradiation, the intensity of the diffraction peaks has become significantly weaker. In general, the weakening of the intensity of the diffraction peaks is the result of poor crystallinity or smaller particle size. But the BET surface area of the fresh-Pt/CeO₂ and laser-irradiated Pt/CeO₂ was about 52.7 and 46.1 m² g⁻¹, respectively (Supplementary Fig. 4a). The corresponding Raman spectra shows a significant enhancement of the characteristic peaks corresponding to the symmetric breathing mode of Ce–O in fluorite CeO₂ after laser treatment[34,35] (Supplementary Fig. 4b). So the decrease of intensity of diffraction peaks can be attributed to the more inferior crystallinity of catalyst, and corresponding HRTEM images can confirm this (Supplementary Fig. 1). The diffraction peak center around 40° can be indexed to the (111) plane of face-centered cubic Pt, which shifted to a higher angle after laser irradiation. Previous reports suggested that this may be due to the incorporation of cerium into the platinum crystal structure to form an alloy, causing lattice contraction[36,37]. In our case, comparing the XPS spectra of Ce 3d before and after laser irradiation, there were no characteristic peaks related to metallic Ce, which indicates that no CePt alloy phase was formed (Supplementary Fig. 5). To demonstrate it more closely, several large size Pt NPs with thick CeOₓ overlays were selected for HRTEM analysis (Supplementary Fig. 6). No CePt alloy phase was observed at the interface between the Pt and CeOₓ layers, which was in agreement with the XPS results. It is worth noting that the surface of Pt NPs is enriched with lattice defects after laser irradiation. Therefore, the shift of the diffraction peak may be due to the distortion of the Pt NP lattice by laser irradiation. The particle size distribution results show the size of Pt NPs did not decrease after the laser treatment and there is no broadening of the diffraction peak. It is known that under ultrafast laser irradiation, abundant metastable structures could be formed in metal NPs because of the strong quenching effect[22]. The weakening of the diffraction peaks can be attributed to the surface-induced poor crystallinity[34]. The formation of a CeOₓ overlayer on the surface of Pt particles is responsible for the seemingly contradictory phenomenon that CeO₂ particles grow after laser irradiation while Pt particles do not show excessive growth. X-ray photoelectron spectroscopy (XPS) was performed to analyze the valence and surface chemical information of the catalyst. The Pt 4f spectra showed that Pt existed mainly in the metallic state before and after laser irradiation with an asymmetric 4f₇/₂ peak centered at 71.05 eV. The peak located at the 72.05 eV is attributed to Pt^δ+, and the intensity of this characteristic peak increases after laser irradiation. As mentioned, Pt/CeO₂ is first dispersed in water and then laser irradiated, and the whole process is exposed to air. On the one hand, water has a certain solubility of oxygen (the solubility of oxygen at 20 °C is 9.17 mg L⁻¹). On the other hand, laser irradiation is performed with continuous agitation, suggesting that the catalysts are all directly exposed to air. In the presence of oxygen, the local heat generated by laser irradiation may induce Pt NPs to bind to oxygen. Moreover, previous studies have shown that in the presence of oxygen involved in the construction of SMSI, Pt–O bonds are generated at the interface between Pt NPs and the overlayer[38,39]. Accordingly, the increase in the intensity of the Pt^δ+ characteristic peak can either be attributed to the partial oxidation of Pt or the formation of Pt–O bonds at the interface between Pt NPs and CeOₓ overlayer. In particular, the signal of the Pt characteristic peak is obviously weakened after laser irradiation. It is known that XPS is a surface analysis technique. After laser treatment, the Pt content decreased from 1.3 to 0.94 atom%, while the Ce content changed from 17.12 to 17.94 atom%. It means that the Pt/Ce ratio on the surface decreases with a CeOₓ overlayer on the Pt surface, which was in good agreement with XRD results.

The suppression of small molecules (such as CO, H₂) adsorption is a typical feature of the classic SMSI. In situ diffuse reflectance infrared Fourier transform spectroscopy (DRIFTS) of CO was employed to explore the adsorption characteristics and electronic structure of the Pt surface at room temperature, and corresponding results are shown in Fig. 3f. Three obvious bands centered at 2094, 2069, and 1836 cm⁻¹ on fresh-Pt/CeO₂, which ascribed to CO linearly adsorbed on ionic Pt (CO-Pt^δ+), metallic Pt (CO-Pt⁰), and bridged CO adsorption on Pt species, respectively[40–42]. After laser irradiation for 1 h, the adsorption of CO molecules on Pt species was significantly weakened. It is already known from the Pt NP size distribution analysis that laser irradiation does not result in significant growth of Pt NPs.

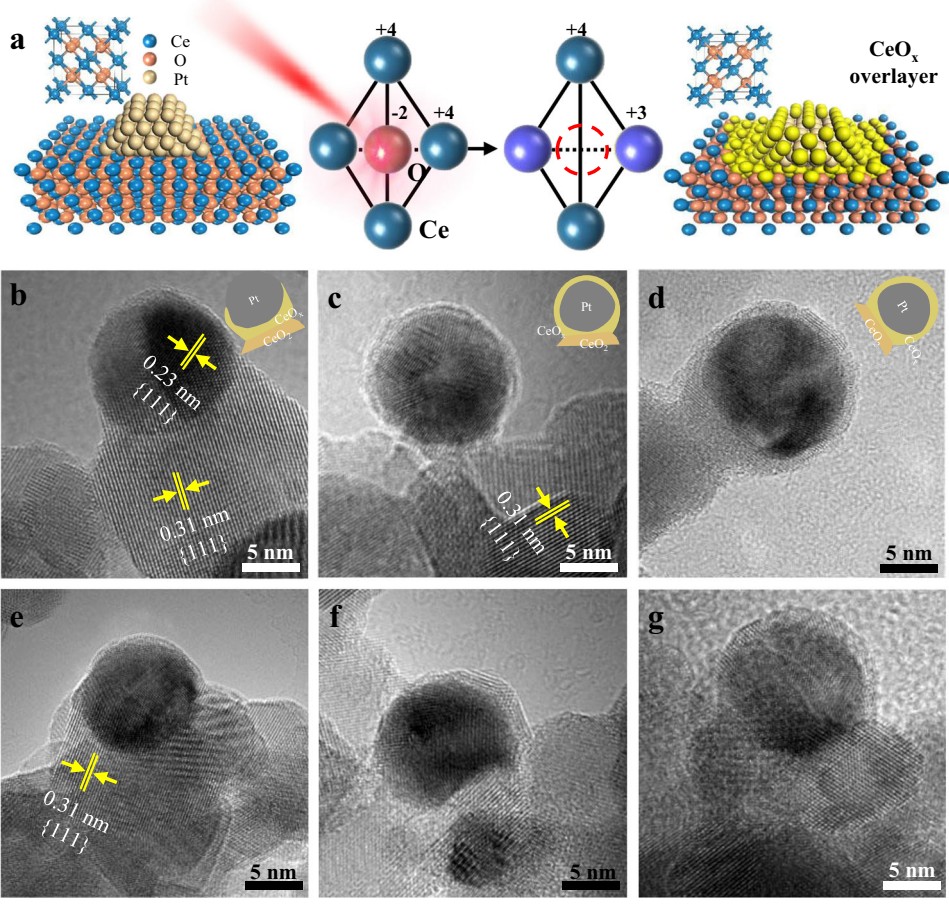

**Fig. 4 Mechanisms of the ultrafast laser-induced SMSI. a** Explanation of ultrafast laser-induced formation of CeO$_x$ overlayer. TEM images of Pt/CeO$_2$ after ultrafast laser irradiation with various exposure time of (**b**) 20 min, **c** 40 min, **d** 60 min (laser power: 250 mW), and laser power of (**e**) 80 mW, **f** 160 mW, **g** 250 mW (exposure time: 60 min). Insets are schematics of laser-fabricated Pt/CeO$_2$ NPs.

Therefore, the suppression of CO adsorption could be reasonably assigned to the coverage of Pt particles by the CeO$_x$ overlayer after laser irradiation, precisely as anticipated for classical SMSI. However, it is noteworthy that the adsorption of CO on Pt species is not completely inhibited, perhaps due to the fact that the CeO$_x$ overlayer is not completely dense but only partially encapsulated, as observed from TEM (Supplementary Fig. 7). Generally, classic SMSI is reversible upon reversal treatment. During the pulsed laser processing, the oxide could be reduced or oxidized, which depended on the initial oxidation state[43]. Under the experimental condition, the CeO$_2$ support was reduced with the increase in Ce$^{3+}$ concentration. The adsorption characteristics and microstructure evolution of laser-Pt/CeO$_2$ under oxidizing atmosphere at 600 °C were investigated. The CO adsorption band was recovered for laser-Pt/CeO$_2$-600 °C-O$_2$ and the corresponding HRTEM showed that the CeO$_x$ overlayer was fade after high-temperature oxidation (Supplementary Fig. 8a, b). This result suggests that laser-induced construction of SMSI was well consistent with classic SMSI. Moreover, the SMSI effect was restored when the laser-Pt/CeO$_2$−600 °C-O$_2$ sample was irradiated with the laser again (Supplementary Fig. 8c–e).

**Mechanisms of the ultrafast laser-induced SMSI.** Several Pt/CeO$_2$ NPs were characterized with increasing exposure time to quantify the ultrafast laser-induced structural reorganization, while the laser power was held constant. As we know, an ultrafast laser with ultrashort pulse duration (<50 fs) and ultrahigh

intensity (>10$^{13}$ W cm$^{-2}$) can almost ionize any materials[44]. Figure 4a shows the laser-induced transformation of CeO$_2$ to CeO$_x$. When the ultrafast laser irradiated on the Pt/CeO$_2$ NPs, the electric field would be confined in the localized Pt/CeO$_2$ interface and enhanced, which arises from the localized plasmon resonance. The enhanced field can induce the nonlinear effects and ionize CeO$_2$. Specifically, when a flux of photons (1.55 eV at 800 nm) was injected, the bounded electrons of CeO$_2$ were excited to the conduction band by multiphoton absorption, leaving the holes in the valence band. On the surface of CeO$_2$, O atoms donated the electrons to the Ce atoms. Therefore, Ce$^{4+}$ accepted the electron to form the Ce$^{3+}$, and the O atoms could be peeled off from the surface of the CeO$_2$ to form the oxygen vacancies[27]. In the crystal structure, two of the cerium ions are replaced by trivalent ions, between which an oxygen vacancy appears[45]. On the other hand, ultrafast photoexcitation of H$_2$O molecules can provide abundant electrons, which can potentially be injected into CeO$_2$ to reduce Ce$^{4+}$. Based on the above analysis, surface defects (Ce$^{3+}$/oxygen vacancy) can be efficiently induced in the interface between the Pt NPs and the CeO$_2$ support where the electric field intensity was larger, which was also demonstrated in the EPR and XPS spectral analysis (Supplementary Fig. 5). In the EPR spectra, compare to fresh Pt/CeO$_2$, the signal of oxygen vacancy ($g = 2.01$) was increased in laser-irradiated Pt/CeO$_2$. CeO$_x$ species may migrate randomly on the unstable surface to form reorganized structures at the perimeter interface[46], and Pt NPs were partially encapsulated (Fig. 4b). The encapsulation that occurred during SMSI can be considered as a

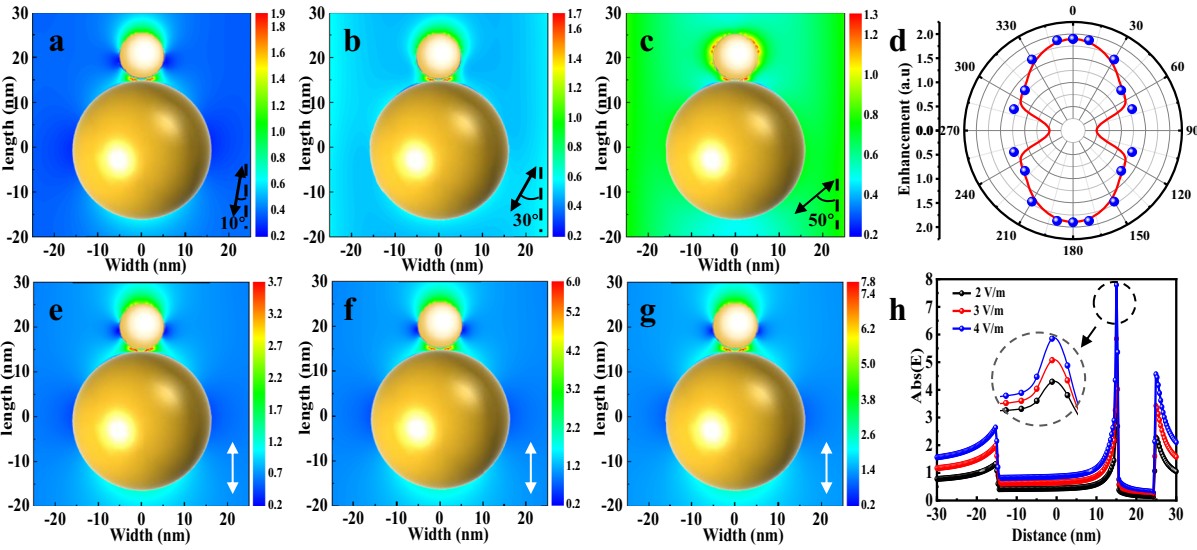

**Fig. 5 Calculated electric field distribution of Pt/CeO$_2$ irradiated with 800-nm laser pulse.** Polarization angles of (**a**) 10°, **b** 30°, and (**c**) 50°. **d** Simulated dependence of the enhanced electric field at the interface between Pt and CeO$_2$ on the polarization angles. Calculated electric field distribution of Pt/CeO$_2$ irradiated with laser intensities of (**e**) 2 V m$^{-1}$, **f** 3 V m$^{-1}$, **g** 4 V m$^{-1}$. **h** The electric field along the longitudinal axis of the Pt/CeO$_2$ excited at 800-nm wavelength. Inset is the enlarged image of the electric field at the interface between Pt and CeO$_2$.

wetting process of the metal NPs by reduced metal oxide. According to the previous reports, the encapsulation process is mainly determined by the surface tension, and larger surface tension of the metal than that of metal oxide support is expected[47]. The case Pt (2.54 J m$^{-2}$) being wetted by CeO$_2$ (1–1.4 J m$^{-2}$) is possible owing to the higher surface tension[48]. The minimization of surface free energy of Pt NPs is the major driven force. According to the XPS results (Supplementary Fig. 9), the Pt/Ce ratio on the surface of specimens were 0.076 (fresh-Pt/CeO$_2$), 0.0459 (Pt/CeO$_2$-laser-20 min), 0.0394 (Pt/CeO$_2$-laser-40 min), 0.0344 (Pt/CeO$_2$-laser-60 min), and 0.0557 (Pt/CeO$_2$-H$_2$), respectively. It means that Pt NPs were more prone to be encapsulated with the increased exposure time[49], and the overlayers become thicker (Fig. 4c, d). Furthermore, several overlayers exhibited lattice fringes that can be identified as CeO$_2$ structures that were epitaxial with the support (Supplementary Fig. 10). When increasing the laser power, the increased thickness of the overlayer was observed (Fig. 4e–g). It means that Pt NPs were more prone to be encapsulated with increasing the deposited energy. To further explain the mechanism of laser-induced SMSI, a reference experiment where CeO$_2$ was irradiated without Pt NPs was performed. According to the XPS results (Supplementary Fig. 11), there were no more surface defects (Ce$^{3+}$/oxygen vacancies) in the laser-treated CeO$_2$. It means that the same laser fluence irradiation without the enhanced electric field could not induce surface defects, and the localized electric field plays a vital role in the formation of surface defects. After that, Pt NPs were loaded on the laser-treated CeO$_2$ and subjected to heat treatment (under argon atmosphere at 500 °C for 2 h). According to the TEM images (Supplementary Fig. 12), no similar overlayers were observed, suggesting that only thermal excitation could not induce SMSI. During the pulsed laser irradiation, the nano-confined field essentially boosts the formation of Ce$^{3+}$/oxygen vacancies and metastable CeO$_x$ migration. On the other hand, thermal excitation could improve the metastable CeO$_x$ mobility.

Pt/CeO$_2$ NPs were dispersed in the liquid, and the angles between laser polarization and the longitudinal axis of Pt/CeO$_2$ NPs were various. In order to analyze the effect of polarization angles on the deposited energy, the electric field distributions of Pt/CeO$_2$ NPs irradiated with different polarization angles at 800-nm wavelength were calculated using finite-difference time-domain (FDTD) simulation. Figure 5a–c shows the electric field distributions excited at polarization angles of 10°, 30°, and 50°, respectively. When the polarization angle was less than 50°, the electric field intensity at the interface between Pt and CeO$_2$ NPs was significantly larger than that on other regions. The dependence of the enhanced electric field at the interface on the polarization angles is given in Fig. 5d. When the polarization angle increased to 90°, no enhanced electric field was observed at the interface. It means that not all polarization angles irradiation can produce enhanced field at the interface. Under the experimental condition with rapid stirring, almost all the NPs can absorb the same amount of energy, which was demonstrated by the previous report[50]. The range of the polarization angles that can result in the enhanced field at the interface was larger, suggesting that ultrafast laser irradiation can efficiently induce structural reorganization. Figure 5e–g shows the electric field distributions excited with laser intensities of 2, 3, and 4 V m$^{-1}$, respectively. The electric field at the interface between Pt and CeO$_2$ increased with the increase of the laser intensity (Fig. 5h), which was consistent with the experimental results. The effect of laser wavelength on the deposited energy was also investigated. Supplementary Figure 13 shows the electric field distributions excited at 400-nm wavelength. Different from 800-nm wavelength excitation, the enhanced field at the interface was observed even when the polarization angle increased to 60°. It should be noted that the enhanced localized fields were dependent on the laser polarization and wavelength, suggesting that the SMSI can be successfully induced in Pt/CeO$_2$ NPs using the proper processing strategy. In order to investigate the effect of the shape of the support, the Pt/CeO$_2$ nanorods (NRs) were synthesized (Supplementary Fig. 14), and the electric field distributions of Pt/CeO$_2$ NRs irradiated with a laser pulse were calculated (Supplementary Fig. 15). The electric field intensity at the interface between Pt and CeO$_2$ NRs was significantly larger than that on other regions. After pulsed laser irradiation, SMSI was successfully fabricated in Pt/CeO$_2$ NRs (Supplementary Fig. 16). In addition, the laser-induced SMSI was also obtained in Pt/TiO$_2$, Pd/TiO$_2$, Au/TiO$_2$, Pt/Al$_2$O$_3$, Pt/SiO$_2$, and Au/MgO NPs, suggesting that this strategy can be extended to other metal/

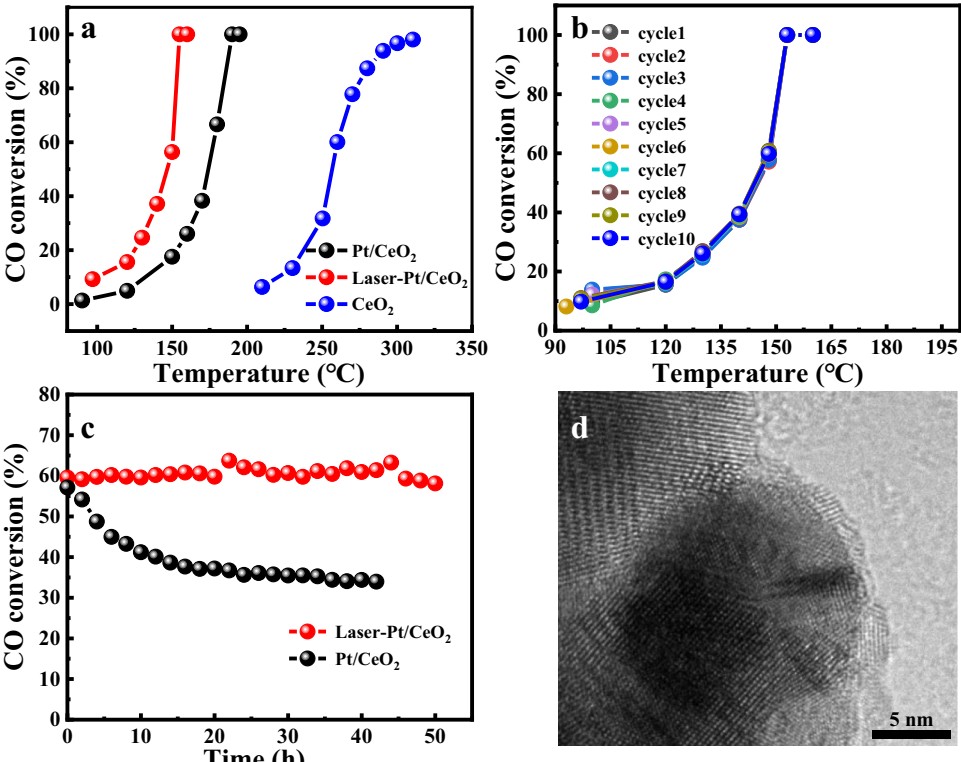

**Fig. 6 Evaluation of Pt/CeO₂ nanocatalysts in CO oxidation. a** CO oxidation curves of laser-irradiated Pt/CeO₂ catalyst and the reference fresh Pt/CeO₂ catalyst. **b** Conversion of CO from 90 to 200 °C with 1st–10th cycles on laser-Pt/CeO₂ catalyst. **c** On-stream reaction of laser-irradiated Pt/CeO₂ and Pt/CeO₂ at 150 °C (space velocity: laser-irradiated Pt/CeO₂ 60,000 mL g⁻¹ h⁻¹, Pt/CeO₂ 20,000 mL g⁻¹ h⁻¹). **d** HRTEM image of laser-irradiated Pt/CeO₂ after CO emission control reaction at 150 °C for 50 h.

metal oxide materials (Supplementary Figs. 17–22 and Supplementary Table 1).

**Catalytic application.** CO oxidation reaction employed as a model reaction to assess the catalytic activity of the catalyst because of its importance in basic research, especially its size-dependent behavior. The T50 (temperature at which 50% conversion of CO is achieved) value is used as an appraisal standard to evaluate the catalytic activity of the catalyst. Figure 6a shows the CO conversion curve of the samples and the catalytic activity of the catalysts decreases in the following sequence: laser-irradiated Pt/CeO₂ (T50 = 145 °C), fresh-Pt/CeO₂ (T50 = 175 °C), CeO₂ (T50 = 250 °C). Obviously, the catalytic activity of laser-irradiated Pt/CeO₂ is significantly superior to that of fresh-Pt/CeO₂, with complete CO conversion at 155 °C for laser-irradiated Pt/CeO₂ and 185 °C for the corresponding fresh-Pt/CeO₂. The apparent activation energy of the reaction obtained by the Arrhenius equation is shown in Supplementary Fig. 23. The apparent activation energy of the catalyst did not change significantly before and after laser irradiation, which were 59.34 and 58.99 kJ mol⁻¹, respectively. The similarity of the apparent activation energy suggests that the catalysts before and after laser treatment follow similar reaction pathways. Therefore, the difference in activity probably mainly derived from the different number of effective active sites rather than the exposed sites, which is in good agreement with previous studies[51–54]. It is generally accepted that CeO₂-loaded Pt NPs in CO catalytic oxidation follow Mars-Van-Krevelen mechanism and that the catalytic activity is size-dependent, where the perimeter of interface between the Pt NPs and CeO₂ is the active site of the reaction[55–57]. This mechanism also worked in our case, in which the improved catalytic activity of the laser-treated sample can be

attributed to the increase in the number of Pt-CeO₂ interfacial sites. According to the above, the size of Pt particles did not decrease after laser irradiation. Thus, the increase in the number of metal-support interface sites was clearly not assigned to a decrease in the size of Pt NPs. Therefore, it is reasonable to infer that the increase in catalytic activity is due to the formation of the SMSI effect. It is already known from the substantial analysis above that the modification of CeOₓ overlayer on the surface of Pt NPs will undoubtedly increase the interface between metal Pt and CeOₓ. However, in general, the catalytic activity of the catalyst tends to degrade after the occurrence of the SMSI effect, due to the presence of overlayer on the metal surface. An important factor resulting in this catalyst passivation is that the overlayer is often dense and thus almost completely obscures the active sites of the catalyst[58]. But in our case, it has been known from the HRTEM representation that laser-induced formation of CeOₓ overlayer is more distinctly different from the classical hydrogen reduction method. Due to the high-energy nature of the femtosecond laser, the overlayer induced with the laser is frequently discontinuous, and even a small number of Pt NPs are only partially encapsulated (Supplementary Fig. 7). The in situ CO-DRIFT and H₂-TPD spectra demonstrate the porous nature of the CeOₓ overlayer (Supplementary Fig. 24). When the femtosecond laser successfully constructs the SMSI effect between Pt and CeO₂, on the one hand, the discontinuous overlayer obscures the sites on the Pt surface, and on the other hand, it provides more active sites at the metal-support interface that are in direct contact with the reactants. This may be the main reason why the catalytic activity of Pt/CeO₂ increases rather than decays after the laser-induced SMSI effect.

To further demonstrate the structural specificity of the laser-induced SMSI effect, the classical method of hydrogen reduction

was also adopted to construct SMSI on $Pt/CeO_2$. According to a previous report, fresh-$Pt/CeO_2$ was calcined at 700 °C under hydrogen atmosphere for 2 h[59]. The HRTEM images revealed that an amorphous and dense $CeO_x$ overlayer was formed on the surface of Pt NPs after $H_2$ reduction, which is more significantly different from the non-dense overlayer formed by laser-induced SMSI, but is consistent with previous reports (Supplementary Fig. 25)[60]. The in situ CO-DRIFT shown that the adsorption of CO is suppressed after high temperature reduction of $Pt/CeO_2$ by hydrogen (Supplementary Fig. 26). The above results all demonstrate the formation of SMSI between Pt and $CeO_2$ after high temperature reduction. CO oxidation was also used as a probe reaction to study the changes in its activity before and after the formation of SMSI. It was found that the catalytic activity of $Pt/CeO_2$ showed a significant decay after the formation of SMSI under hydrogen reduction, and its complete conversion temperature increased from 170 to 230 °C for fresh samples. However, the catalytic activity was restored after reoxidation of the $H_2$-$Pt/CeO_2$ sample under air atmosphere at 600 °C, and its complete conversion of CO was achieved at 160 °C (Supplementary Fig. 27). The above experimental phenomena are in good agreement with the uniform classical SMSI, but differ significantly from the laser-induced SMSI effect. The root cause of this difference is perhaps the difference in the structure of the $CeO_x$ overlayer, as mentioned above. Although the $H_2$ reduction construct SMSI effect is also able to increase the interface between the Pt and $CeO_x$, but the dense $CeO_x$ overlayer formed on the surface of Pt NPs makes it difficult for the reactants to directly contact the interfacial sites, which instead reduces the activity of the catalyst. The laser-irradiated $Pt/CeO_2$ exhibits excellent cycling stability due to the SMSI effect, with no decay in activity after ten cycles (Fig. 6b). It is generally accepted that the low on-stream stability of supported noble metal catalysts has been a major challenge preventing their practical applications. Whereas the previous studies have shown that constructing SMSI effects can significantly improve the stability of supported noble metal catalysts[61,62]. Therefore, we investigated the stability of laser-irradiated $Pt/CeO_2$ for CO catalytic oxidation. As expected, laser-irradiated $Pt/CeO_2$ showed no significant decay in activity after 50 h of reaction at 150 °C, while the corresponding fresh-$Pt/CeO_2$ decreased gradually (Fig. 6c). HRTEM was performed to analyze the microstructure of laser-irradiated $Pt/CeO_2$ after reaction (Fig. 6d). The overlayer still kept intact with no apparent fade even after prolonged oxidation, which demonstrates the high reliability.

## Discussion

In summary, we proposed a novel approach to induce SMSI in $CeO_2$-supported Pt NPs based on ultrafast laser excitation. It is found that the electric field intensity at the interface between Pt and $CeO_2$ was significantly larger than that on other regions. The surface defects and metastable $CeO_x$ migration were formed under the local field excitation, indicating that the nanoconfined electric field played a key role in the formation process of SMSI. The ultrafast laser-induced SMSI with porous overlayers possessed more effective active sites and exhibited superior catalytic activity and stability. We also used this strategy to induce SMSI in $Pt/TiO_2$, $Pd/TiO_2$, $Pt/Al_2O_3$, $Pt/SiO_2$, $Au/TiO_2$, and $Au/MgO$ NPs, showing its possibility of extending to other metal/metal oxide materials. We suggest that this study provides new insights for the formation of SMSI, and opens a general pathway to create novel nanomaterials, which have promising applications.

## Methods

**Raw materials**. All the chemical reagent was analytically pure and without further refinement before use. $CeO_2$ NPs were purchased from Beijing DK nano S&T Ltd,

with the purity of 99.99% and the specific surface area of 30–50 $m^2 g^{-1}$. Chloroplatinic acid ($H_2PtCl_4$, AR, 99.99%) was purchased from Sigma Aldrich. $NaHB_4$ (AR, 99.9%) was purchased from Sinopharm Chemical Reagent Beijing Co., Ltd.

**Preparation of fresh-$Pt/CeO_2$**. In all, 0.2 g $CeO_2$ NPs were dispersed in 180 mL deionized water under ultra-sonication to obtain a homogenous slurry. Then 4 mL $H_2PtCl_4$ (5 mmol $L^{-1}$) solution was added into the slurry under stirring at room temperature for 30 min. Afterward, 4 mL newly prepared $NaBH_4$ solution (0.01 mol $L^{-1}$) was dropped into the slurry under rapid stirring (800 r $min^{-1}$) for 1 h. The sample was washed with an amount of deionized water, until no obvious precipitation can be seen in wash water tested with $AgNO_3$. After separation and drying, the resulting $Pt/CeO_2$ catalyst was calcined in a glass tube furnace under argon atmosphere at 500 °C for 2 h to obtain fresh-$Pt/CeO_2$.

**Preparation of laser-irradiated $Pt/CeO_2$**. Femtosecond laser pulses (35 fs) emitted by an amplified Ti-Sapphire system (central wavelength: 800 nm, repetition rate: 1 kHz) was used as an excitation source to induce nanoconfined electric field. A frequency doubling crystal was employed to generate laser pulses centered at 400 nm. The temporal profile of the pulses was diagnosed by second harmonic autocorrelation, whereas laser fluence (80–250 mW) control was performed by a half-wave plate and a linear polarized beam splitter. The exposure time (20–60 min) was controlled using a shutter connected to a computer. A six-freedom translation platform was used to ensure a proper excitation of the entire solution. Generally, 50 mg $Pt/CeO_2$ powders were dispersed in 2 mL deionized water, followed by ultrasonic stirring for 30 min. The laser pulses were focused below the air–liquid interface using a lens, and the incident spot size was fixed at about 5 mm. After homogeneous processing, laser-induced SMSI were produced. Then, 2 mL deionized water was added, and the nanostructures were dried at 50 °C for further characterization and surface activity tests.

**Structural characterizations**. The morphology, microstructure, and EDS-Mapping of the samples were characterized by a JEOL JEM-1010/2010 transmission electron microscope operating at 200 kV. X-ray diffraction (XRD) patterns were performed on a Bruker D8-Advance diffractometer using Ni-filtered Kα radiation. EELS spectra and corresponding HAADF-STEM images were conducted on Titan cubed Themis G2300 double aberration-corrected TEM equipped with a Quantum ER965 type EELS accessory. BET-specific surface area measurements were measured on an Autosorb-iQ2-MP automated gas sorption system. X-ray photoelectron spectra (ESCALAB 250Xi) were employed to analyze the valence states of the sample. EPR spectra were obtained at 77 K with a Bruker A300-10/12. $H_2$-TPD spectra was obtained with a chemisorption apparatus (Autochemll2920). In situ DRIFTS spectra were obtained with a Bruker v70 spectrometer equipped with a mercury telluride detector with a resolution of 4 $cm^{-1}$. Twenty millgrams sample was placed in a stainless steel crucible and then loaded into a ZnSe window that could be operated at high temperatures. Before the adsorption of CO, the samples were pre-treated in situ for 1 h in a He gas stream (33.3 mL $min^{-1}$) at 120 °C and then cooled to room temperature. The gas stream was switched to pure He to collect background spectra. Subsequently, a mixture of 3 vol% CO/He (33.3 mL $min^{-1}$) was introduced into the reaction cell, after the CO adsorption saturation, the gaseous CO was flashed off by evacuation and then the spectra were collected. For reoxidation samples, in situ heating at 10 vol% $O_2$/He (33.3 mL $min^{-1}$) at 600 °C for 1 h was also required before the background spectra were collected. For $H_2$-$Pt/CeO_2$, fresh-$Pt/CeO_2$ was calcined in $H_2$ atmosphere at 700 °C for 2 h and then the spectra were collected by the above method.

**CO oxidation**. The catalytic activity and stability of the sample were evaluated by a fixed-bed flow microreactor equipped with Shimadzu GC-2014 gas chromatograph under atmospheric pressure. Typically 50 mg catalyst mixed with 150 mg quartz sands to prevent reaction runaway. Then, the mixture was transferred into a quartz tube and secure with a quartz face. The composition of the reaction gas was CO/ $O_2$/$N_2$ (1:20:79), and the reaction gas hourly space velocity was 60,000 mL $g^{-1} h^{-1}$.

## Data availability

The data that support the findings of this study are available within the paper and its Supplementary Information, and all data are available from the authors on reasonable request.

## References

1.  van Deelen, T. W., Mejía, C. H. & de Jong, K. P. Control of metal-support interactions in heterogeneous catalysts to enhance activity and selectivity. *Nat. Catal.* **2**, 955–970 (2019).

# ARTICLE

2. Li, Z. et al. Well-defined materials for heterogeneous catalysis: from nanoparticles to isolated single-atom sites. *Chem. Rev.* **120**, 623–682 (2019).

3. Mitchell, S., Qin, R., Zheng, N. & Pérez-Ramírez, J. Nanoscale engineering of catalytic materials for sustainable technologies. *Nat. Nanotechnol.*, **16**, 1–11 (2020).

4. Tauster, S., Fung, S. & Garten, R. L. Strong metal-support interactions. Group 8 noble metals supported on titanium dioxide. *J. Am. Chem. Soc.* **100**, 170–175 (1978).

5. Tauster, S., Fung, S., Baker, R. & Horsley, J. Strong interactions in supported-metal catalysts. *Science* **211**, 1121–1125 (1981).

6. Li, Z. et al. Reactive metal–support interactions at moderate temperature in two-dimensional niobium-carbide-supported platinum catalysts. *Nat. Catal.* **1**, 349–355 (2018).

7. Belzunegui, J. P., Sanz, J. & Rojo, J. M. Contribution of physical blocking and electronic effect to establishment of strong metal-support interaction in rhodium/titanium dioxide catalysts. *J. Am. Chem. Soc.* **114**, 6749–6754 (1992).

8. Ko, C. & Gorte, R. A comparison of titania overlayers on Pt, Pd and Rh. *Surf. Sci.* **161**, 597–607 (1985).

9. Zhang, S. et al. Dynamical observation and detailed description of catalysts under strong metal–support interaction. *Nano Lett.* **16**, 4528–4534 (2016).

10. Tang, H. et al. Classical strong metal–support interactions between gold nanoparticles and titanium dioxide. *Sci. Adv.* **3**, 1700231 (2017).

11. Wang, L., Wang, L., Meng, X. & Xiao, F. S. New strategies for the preparation of sinter-resistant metal-nanoparticle-based catalysts. *Adv. Mater.* **31**, 1901905 (2019).

12. Qin, Z.-H., Lewandowski, M., Sun, Y.-N., Shaikhutdinov, S. & Freund, H.-J. Encapsulation of Pt nanoparticles as a result of strong metal−support interaction with Fe₃O₄ (111). *J. Phys. Chem. C* **112**, 10209–10213 (2008).

13. Du, X. et al. Size-dependent strong metal-support interaction in TiO₂ supported Au nanocatalysts. *Nat. Commun.* **11**, 1–8 (2020).

14. Liu, J. et al. Deep understanding of strong metal interface confinement: a journey of Pd/FeOₓ catalysts. *ACS Catal.* **10**, 8950–8959 (2020).

15. Wang, H. et al. Strong metal–support interactions on gold nanoparticle catalysts achieved through Le Chatelier's principle. *Nat. Catal.* **4**, 418–424 (2021).

16. Liu, S. et al. Ultrastable Au nanoparticles on titania through an encapsulation strategy under oxidative atmosphere. *Nat. Commun.* **10**, 1–9 (2019).

17. Zhang, J. et al. Wet-chemistry strong metal–support interactions in titania-supported Au catalysts. *J. Am. Chem. Soc.* **141**, 2975–2983 (2019).

18. Zhu, D., Yan, J., Xie, J., Liang, Z. & Bai, H. Ultrafast laser-induced atomic structure transformation of Au nanoparticles with improved surface activity. *ACS Nano* **15**, 13140–13147 (2021).

19. Zhang, D., Gökce, B. & Barcikowski, S. Laser synthesis and processing of colloids: fundamentals and applications. *Chem. Rev.* **117**, 3990–4103 (2017).

20. Zhu, D., Yan, J., Liang, Z., Xie, J. & Bai, H. Laser stripping of Ag shell from Au@Ag nanoparticles. *Rare Met.* **40**, 3454–3459 (2021).

21. Li, Z. et al. A silver catalyst activated by stacking faults for the hydrogen evolution reaction. *Nat. Catal.* **2**, 1107–1114 (2019).

22. Yan, J., Zhu, D., Xie, J., Shao, Y. & Xiao, W. Light tailoring of internal atomic structure of gold nanorods. *Small* **16**, 2001101 (2020).

23. Chen, C. H. et al. Ruthenium-based single-atom alloy with high electrocatalytic activity for hydrogen evolution. *Adv. Energy Mater.* **9**, 1803913 (2019).

24. Zhu, D., Yan, J. & Xie, J. Reshaping enhancement of gold nanorods by femtosecond double-pulse laser. *Opt. Lett.* **45**, 1758 (2020).

25. Körstgens, V. et al. Laser-ablated titania nanoparticles for aqueous processed hybrid solar cells. *Nanoscale* **7**, 2900–2904 (2015).

26. Li, L., Yu, L., Lin, Z. & Yang, G. Reduced TiO₂-graphene oxide heterostructure as broad spectrum-driven efficient water-splitting photocatalysts. *ACS Appl. Mater. Interfaces* **8**, 8536–8545 (2016).

27. Chen, X. et al. Laser-modified black titanium oxide nanospheres and their photocatalytic activities under visible light. *ACS Appl Mater. Interfaces* **7**, 16070–16077 (2015).

28. Tian, F. et al. Preparation and photocatalytic properties of mixed-phase titania nanospheres by laser ablation. *Mater. Lett.* **63**, 2384–2386 (2009).

29. Pyatenko, A., Wang, H. & Koshizaki, N. Growth mechanism of monodisperse spherical particles under nanosecond pulsed laser irradiation. *J. Phys. Chem. C* **118**, 4495–4500 (2014).

30. Lau, M., Reichenberger, S., Haxhiaj, I., Barcikowski, S. & Müller, A. M. Mechanism of laser-induced bulk and surface defect generation in ZnO and TiO₂ nanoparticles: effect on photoelectrochemical performance. *ACS Appl. Energy Mater.* **1**, 5366–5385 (2018).

31. Amendola, V. et al. Room-temperature laser synthesis in liquid of oxide, metal-oxide core-shells, and doped oxide nanoparticles. *Chemistry* **26**, 9206–9242 (2020).

32. Zhu, L. et al. Visualizing anisotropic oxygen diffusion in Ceria under activated conditions. *Phys. Rev. Lett.* **124**, 056002 (2020).

33. Douillard, L. et al. Local electronic structure of Ce-doped Y₂O₃: an XPS and XAS study. *Phys. Rev. B* **49**, 16171 (1994).

34. Spanier, J. E., Robinson, R. D., Zhang, F., Chan, S.-W. & Herman, I. P. Size-dependent properties of CeO2- y nanoparticles as studied by Raman scattering. *Phys. Rev. B* **64**, 245407 (2001).

35. Hattori, T., Kobayashi, K. & Ozawa, M. Size effect of Raman scattering on CeO₂ nanocrystal by hydrothermal method. *Jpn J. Appl. Phys.* **56**, 01AE06 (2016).

36. Beck, A. et al. The dynamics of overlayer formation on catalyst nanoparticles and strong metal-support interaction. *Nat. Commun.* **11**, 1–8 (2020).

37. Hardacre, C., Rayment, T. & Lambert, R. M. Platinum/ceria CO oxidation catalysts derived from Pt/Ce crystalline alloy precursors. *J. Catal.* **158**, 102–108 (1996).

38. Wu, Z., Li, Y. & Huang, W. Size-dependent Pt-TiO₂ strong metal–support interaction. *J. Phys. Chem. Lett.* **11**, 4603–4607 (2020).

39. Macino, M. et al. Tuning of catalytic sites in Pt/TiO2 catalysts for the chemoselective hydrogenation of 3-nitrostyrene. *Nat. Catal.* **2**, 873–881 (2019).

40. Ding, K. et al. Identification of active sites in CO oxidation and water-gas shift over supported Pt catalysts. *Science* **350**, 189–192 (2015).

41. Nie, L. et al. Activation of surface lattice oxygen in single-atom Pt/CeO2 for low-temperature CO oxidation. *Science* **358**, 1419–1423 (2017).

42. Jones, J. et al. Thermally stable single-atom platinum-on-ceria catalysts via atom trapping. *Science* **353**, 150–154 (2016).

43. Yu, M. et al. Laser fragmentation-induced defect-rich cobalt oxide nanoparticles for electrochemical oxygen evolution reaction. *ChemSusChem* **13**, 520–528 (2020).

44. Shugaev, M. V. et al. Fundamentals of ultrafast laser–material interaction. *MRS Bull.* **41**, 960–968 (2016).

45. Melchionna, M. & Fornasiero, P. The role of ceria-based nanostructured materials in energy applications. *Mater. Today* **17**, 349–357 (2014).

46. Kuwauchi, Y., Yoshida, H., Akita, T., Haruta, M. & Takeda, S. Intrinsic catalytic structure of gold nanoparticles supported on TiO₂. *Angew. Chem. Int. Ed. Engl.* **51**, 7729–7733 (2012).

47. Du, X. et al. Size-dependent strong metal-support interaction in TiO₂ supported Au nanocatalysts. *Nat. Commun.* **11**, 5811 (2020).

48. Hayun, S., Ushakov, S. V., Navrotsky, A. & Klimm, D. Direct measurement of surface energy of CeO₂ by differential scanning calorimetry. *J. Am. Ceram. Soc.* **94**, 3679–3682 (2011).

49. Lau, M., Ziefuss, A., Komossa, T. & Barcikowski, S. Inclusion of supported gold nanoparticles into their semiconductor support. *Phys. Chem. Chem. Phys.* **17**, 29311–29318 (2015).

50. González-Rubio, G., Díaz-Núez, Pablo & Rivera, Antonio Femtosecond laser reshaping yields gold nanorods with ultranarrow. *Science* **358**, 640–644 (2017).

51. Grisel, R. & Nieuwenhuys, B. Selective oxidation of CO, over supported Au catalysts. *J. Catal.* **199**, 48–59 (2001).

52. Ke, J. et al. Strong local coordination structure effects on subnanometer PtOx clusters over CeO₂ nanowires probed by low-temperature CO oxidation. *ACS Catal.* **5**, 5164–5173 (2015).

53. Pacchioni, G. & Freund, H.-J. Controlling the charge state of supported nanoparticles in catalysis: lessons from model systems. *Chem. Soc. Rev.* **47**, 8474–8502 (2018).

54. Farmer, J. A. & Campbell, C. T. Ceria maintains smaller metal catalyst particles by strong metal-support bonding. *Science* **329**, 933–936 (2010).

55. Zhang, B. & Qin, Y. Interface tailoring of heterogeneous catalysts by atomic layer deposition. *ACS Catal.* **8**, 10064–10081 (2018).

56. Farnesi Camellone, M., Negreiros Ribeiro, F., Szabová, L., Tateyama, Y. & Fabris, S. Catalytic proton dynamics at the water/solid interface of ceria-supported Pt clusters. *J. Am. Chem. Soc.* **138**, 11560–11567 (2016).

57. Conner, W. C. Jr & Falconer, J. L. Spillover in heterogeneous catalysis. *Chem. Rev.* **95**, 759–788 (1995).

58. Liu, X. et al. Strong metal–support interactions between gold nanoparticles and ZnO nanorods in CO oxidation. *J. Am. Chem. Soc.* **134**, 10251–10258 (2012).

59. Bernal, S. et al. Nanostructural evolution of a Pt/CeO₂ catalyst reduced at increasing temperatures (473–1223 K): a HREM Study. *J. Catal.* **169**, 510–515 (1997).

60. Bernal, S. et al. Electron microscopy (HREM, EELS) study of the reoxidation conditions for recovery of NM/CeO2 (NM: Rh, Pt) catalysts from decoration or alloying phenomena. *Catal. Lett.* **76**, 131–137 (2001).

61. Zhang, Y. et al. Boosting the catalysis of gold by O₂ activation at Au-SiO₂ interface. *Nat. Commun.* **11**, 1–10 (2020).

62. Ta, N. et al. Stabilized gold nanoparticles on ceria nanorods by strong interfacial anchoring. *J. Am. Chem. Soc.* **134**, 20585–20588 (2012).

## Acknowledgements

This research was supported by the National Natural Science Foundation of China (52173257, 51872159, 51775303, 52075289).

## Author contributions

J.Z. and D.Z. performed the experimental studies and carried out the analysis. All the authors discussed the result and commented on the manuscript. J.Y. and C.-A.W. jointly designed the study and supervised the project.

## Competing interests

The authors declare no competing interests.
