## [Peer Review File · Nature Communications]

Title: Strong metal-support interactions induced by an ultrafast laserREVIEWER COMMENTS

Reviewer #1 (Remarks to the Author):

The present manuscript deals with the laser-induced SMSI for Pt/CeO₂ oxidation catalysts. Laser processing of the Pt/CeO₂ was conducted in water by employing fs-laser pulses emitted by a Ti:sapphire fs-laser. The laser post-processed catalysts clearly show a cerium oxide overlayer grown over the Pt-NP as evidenced by HR-TEM. From XPS and EELS reduced Ce³⁺ species were identified. The weakening of XRD reflexes indicates a higher degree of amorphousness. DRIFTS in the CO atmosphere show the disappearance of CO bands linked to Pt-CO bonds after laser processing further supporting the interpretation of a laser-induced SMSI. Results of the duration of laser processing (= number of laser pulses) and laser intensity show increasing thickness of the cerium oxide overlayer. Results were further verified with frequency-doubled laser pulses and transferability was demonstrated at the example of Pt-TiO₂ which also formed overgrown structures. The laser-treated Pt-CeO_x obtained a superior catalytic CO oxidation activity and superior cycle stability compared to the untreated sample. Since the activation energy of the catalyst didn't change on laser treatment the authors hypothesize that the overgrowth was only partial and active sites are still present. Overall the manuscript is mostly concise and informative addressing a novel method to induce /trigger SMSI in heterogenous catalysts with reducible oxides in a controlled manner. A few questions are yet left open and need to be addressed to create a full picture before final publication can be recommended. The following points should be addressed:

- 1) Please comment on the role of chloride residues in laser-induced overgrowth. Provide evidence (e.g. EELS or XPS) that no chloride was present during laser treatment potentially affecting the overgrowth.
- 2) The authors convincingly discuss that minimization of the surface energy drives the overgrowth. With this interpretation, the minimization of surface energy is the driving force for the process. Yet, to fully understand the laser-induced processes the role of thermal excitation causing an improved atom mobility and laser-induced defect formation (with defects altering the surface energy and hence the driving force) occur at the same time. Hence, the following important question remains: What drives the overgrowth, the laser-induced defect formation in CeO₂, or the thermal excitation? The question is the key to finalize the story and can be addressed with a simple reference experiment where both processes are disentangled:

Please provide a reference experiment where CeO₂ is irradiated without Pt NPs present and add the Pt-NPs after laser treatment. Please characterize the laser-treated cerium oxide by XPS and/or EELS and validate whether Ce³⁺ formed during laser treatment even though Pt-NPs are absent. This reference experiment thereby will also verify the discussed story on the field enhancement (absent in these experiments). Finalize the reference experiment by a respective HR-TEM investigation of the laser-treated CeO₂ after its functionalization with Pt-NPs to see if overgrowth did happen even without thermal excitation (only driven by the additional laser-induced Ce³⁺ defects) or not. Depending on the outcome the key role either of the laser-induced defects or the thermal excitation will be identified as a trigger for the overgrowth.

- 3) The interpretation of a partial overgrowth (brought forward due to the observed independence of the activation energy before and after laser treatment) is not concise with CO DRIFTS where the disappearance of the CO-Pt band was discussed as evidence for the overgrowth. If only partial

overgrowth would be the case, a CO-Pt signal should have remained in DRFITS but was not observed. To remedy this issue; what is the active site for CO oxidation in the case of the Pt/CeO₂ catalyst? Please clearly state in the manuscript what the active sites are for this catalyst. Is it the Pt surface itself (as inferred, yet not directly mentioned by the partial overgrowth interpretation) or is the activity dominated by a Mars-Van-Krevelen mechanism at the interface of Pt and CeO₂ (also compare the observed activation enthalpy with literature values for pure Pt-NPs without CeO₂ to identify the role of CeO₂)? If a M-V-K based mechanism from an interaction between Pt-CeO₂ provides the active sites for CO oxidation the observed independence of the activation enthalpy and disappearance of the Pt-CO band would coincide. In this interpretation, the overgrowth would mainly have increased the contact area between Pt and CeO₂ and hence the number of active sites while the overgrowth would not have changed the nature of the active sites (and hence no change in activation energy). Please revise this aspect in your manuscript and clarify what the active sites are and why the Pt-CO signal disappears after laser treatment although the activation enthalpy remains constant. Partial overgrowth is insufficient to explain both latter observations.

Further minor comments:

4) Line 78-80: "However, the high temperature and high pressure caused by the nanosecond or longer pulse irradiation may induce unfavorable phase transformation with inferior properties²⁶." Please specify. This sentence is too general. What phase transformations do you refer to?

5) Following sentence in line 80-83: "In addition, the noticeable photothermal effects involved in long-pulse laser irradiation might cause sintered NPs, resulting in decreased catalytic activity due to the sacrifice of active sites²⁷." ♦ The sentence and the included statement is potentially misleading as the cited paper deals with laser melting where aggregated NPs are molten together to form larger NPs. Non-aggregated particles will not show this sintering under ns-laser treatment but potential defect formation which is also referred to as laser defect engineering (LDL; pls see the current review of Amendola et al. DOI: 10.1002/chem.202000686).

Understandably, the author's intention in this paragraph was to draw attention to the importance of ultrafast lasers. The biggest advantage of ultrafast laser pulses is the strong electronic excitation (high electron temperature) and (compared to ns-lasers) lesser heat load in the lattice. Consequently, it was found that LDL of TiO₂ with picosecond laser pulses led to 2 times higher photocurrents discussed with the generation of defects when a sufficient number of laser pulses is applied (too high led to segregation and again lower activity). Nanosecond laser pulses in turn directly led to a gradual decrease of photocurrent potentially due to the formation of crystal defects in the lattice due to isochoric melting and subsequent rapid quenching (DOI: 10.1039/c5cp04296h). A more comprehensive review on this can also be found in the recent review of Amendola et al. (DOI: 10.1002/chem.202000686) The passage should be revised and clarified accordingly.

6) Figure 1a: the legend shows Ce, O, and C but no carbon can be seen in the figure. Also, the particle should be denoted as e.g. "Pt NP" for clarity

7) Figure 1b: The figures take some time to understand. To me, it seems more logical to switch the two figures to first show the 'zoomed-out' version and then the 'zoomed-in' version with the local field, etc. Same for Fig. 1c.

8) Line 184-186: "It is well known that XPS is a surface analysis technique and that the Pt/Ce ratio on the surface decreases significantly with a CeOx overlayer on the Pt surface, which was good agreement with XRD results." Where is this Pt/Ce ratio shown? Please provide how the at% of Ce and Pt (determined from XPS) changes before and after laser treatment.

9) Line 197: "...it is also clear that the Pt NPs were not oxidation after laser irradiation." Please check the language.

10) Line 202-204: "Generally, classic SMSI is reversible upon reversal treatment. Our analysis combining with the previous research reveal that the process of laser irradiation is somewhat reductive." Recently, at the example of laser processing of Co₃O₄ and CoO, it was shown (DOI: 10.1002/cssc.201903186) that the initial oxidation state of the oxide affects whether the process is reductive or oxidative. Please consider including this aspect in the manuscript.

11) Line 208-211: "This result suggests that laser-induced construction of SMSI should be irreversible in the oxidizing atmosphere, which was not consistent with classic SMSI and the reasons for this still need to be further studied." Please comment if there is any evidence of Pt-Ce-mixed oxides formed in the overgrown layer.

12) In a previous paper (DOI: 10.1039/c5cp04296h) the inclusion of AuNP in ZnO with an increasing number of laser cycles (= number of laser pulses) well in line with the discussion on irradiation duration and increasing thickness of the overgrown layer and hence could be mentioned in this context.

Reviewer #2 (Remarks to the Author):

The authors report the use of a femtosecond laser to induce strong metal support interactions (SMSI) in a Pt/CeO₂ material (i.e., overcoating of Pt by the suboxide of ceria). It is argued that such SMSI are beneficial for low-temperature oxidation of CO relative to the untreated Pt/CeO₂ that does not contain such SMSI. The manuscript compares characterization data and catalytic activity data for laser-treated and untreated Pt/CeO₂. To understand the relevance of this exotic way to induce SMSI (with a femtosecond laser), it is necessary to provide the respective benchmark data for the Pt/CeO₂ material with conventionally-induced SMSI, i.e. by pretreating Pt/CeO₂ under H₂ prior to the catalytic test (and doing selected characterization). Without this data, the Manuscript appears incomplete and assessment of its importance and impact is difficult. That being said, it well may be that the overcoatings that has been obtained in an aqueous suspension without removal of air using laser (oxidizing environment) and under H₂ flow (reducing environment) are different chemically. Yet it needs to be demonstrated in the first place.

Additional questions (according to the flow of the Manuscript but not their importance) are as follows:

1) Why is it relevant to include "ultrafast" in the title "Ultrafast laser-induced strong metal-support interactions in Pt/CeO₂ with highly catalytic stability"? The recipe to prepare materials takes up to 1 hour, this is hardly ultrafast. "Highly catalytic stability" is confusing.

2) The authors refer to ZnO as a non-reducible oxide. This is not correct, ZnO is regarded as reducible (in contrast to silica, alumina, magnesia, etc).

3) Simple H₂ pretreatment typically used to induce SMSI is presented as requiring "harsh experimental

conditions (high temperature, explosive reductant (H₂))” – this is an overstatement, especially relative to the complexity of the method presented in this work.

4) The authors write that they have developed a material “which possesses a strong quenching effect to generate metastable nanostructures with extraordinary properties.” – it is not clear what extraordinary properties have been discovered in the present work. None of the properties discussed appears extraordinary.

5) Figures 1a and 1b contain a label of carbon, however the structures presented are carbon-free.

6) The intensity of diffraction peaks has decreased after the laser treatment. The authors propose that “The weakening of the diffraction peaks can therefore be explained by the presence of the CeOx overlayer.” This explanation does not look convincing, given the small thickness of the overlayer observed by TEM imaging.

7) The XPS peak at ca. 72.05 eV in Pt/CeO₂ is attributed to Pt^{δ+}. The intensity of this peak is found to increase after laser irradiation, explained by the formation of Pt-O at the interface between Pt and CeOx overlayer. Next, “transferring electrons from Pt to CeOx” is suggested. It has been reported in the literature that the SMSI effect (geometric effect, overcoating) often occurs along with the EMSI effect, i.e. electronic metal support interaction. The latter includes the electron transfer, as mentioned, yet the transfer of electrons in the opposite direction is typical, that is from the reduced CeOx to a supported noble metal such as Pt, Ru, etc. This creates M(δ⁻) states, which are more reduced in XPS than M(0) states. Based on this, the suggested “transferring electrons from Pt to CeOx” may not be correct and requires additional experimental evidence. Can it be that the laser exposure formed small Pt clusters or atoms?

8) The plot with Pt/Ce ratios is missing. This data is essential to demonstrate the occurrence of SMSI (in addition to the TEM data, which is a local method and probes only a fraction of specimen). It is suggested that the authors add this plot for materials of Figure 4, i.e., after 20, 40 and 60 min of laser irradiation. Is it possible to present and discuss Pt/Ce(4⁺) and Pt/Ce(3⁺) ratios for these samples? How would those ratios compare to the reference Pt/CeO₂-H₂ material?

9) IR data with CO probe molecule in Fig. 3f is not suitable because too large amounts of CO have been admitted to the cell, which creates large peaks due to CO gas-phase absorption. This covers area of more blue-shifted CO peaks (up to ca. 2175 cm⁻¹). So it is not possible to evaluate the presence or absence of Pt(δ⁺)-CO adducts. Rather use pulses of small amounts of CO to the outgassed specimen, from ca. 0.1 mbar to ca. 1 mbar.

10) How similar (or dissimilar) are carbonate bands on discussed materials? It can be that carbonates form after oxidation of CO by the lattice oxygen already at room temperature.

11) Did the authors attempt to evaluate the metal surface area in their materials? H₂-TPD can be a method of choice. Having this information will allow to evaluate the suggested porosity of the overlayer and availability of the metal surface. It is advised to provide this data. Likewise, CO-TPD measurement can also be deployed. The porosity of CeOx is suggested but no experimental data directly supporting this speculation has been provided.

12) Figure 4a and respective discussion in the text. Authors are asked to justify (with citation to relevant literature) their assertion that oxygen vacancy site in ceria is necessarily reducing two Ce⁴⁺ ions to two Ce³⁺ ions. Is it correct that this electron density is localized fully at the oxygen vacancy site (and not delocalized), disregard of the density of oxygen vacancy sites? EPR data shows only a minor signal

increase in the laser-treated sample.

13) Cycles of CO oxidation are discussed, but what is a cycle? This has not been defined in the Manuscript or Supplementary Information.

14) Arrhenius plots in Fig. 6b show the difference in E_a for the two catalysts studied of merely 0.3 kJ mol^{-1} . Yet the temperature difference to reach 50% CO conversion with these catalysts is ca. $50 \text{ }^\circ\text{C}$. Does this make sense?

15) Figure S2, y-axis. Replace "probability" by "particle count". Without this information it is not clear if 10 particles have been analyzed or 200 particles.

16) Reporting BET surface area with the precision to the third decimal point (Figure S4) is unreasonable. The error of the measurement is larger than what the authors suggest with their third decimal point accuracy.

17) Can the authors demonstrate SMSI effect for Pt particles on more challenging irreducible supports? Ceria and titania are well known for SMSI effects, this is easily achieved by a simple H_2 pretreatment, therefore current results are not very surprising.

Response to reviewers

Reviewer #1 (Remarks to the Author):

The present manuscript deals with the laser-induced SMSI for Pt/CeO₂ oxidation catalysts. Laser processing of the Pt/CeO₂ was conducted in water by employing fs-laser pulses emitted by a Ti:sapphire fs-laser. The laser post-processed catalysts clearly show a cerium oxide overlayer grown over the Pt-NP as evidenced by HR-TEM. From XPS and EELS reduced Ce³⁺ species were identified. The weakening of XRD reflexes indicates a higher degree of amorphousness. DRIFTS in the CO atmosphere show the disappearance of CO bands linked to Pt-CO bonds after laser processing further supporting the interpretation of a laser-induced SMSI. Results of the duration of laser processing (= number of laser pulses) and laser intensity show increasing thickness of the cerium oxide overlayer. Results were further verified with frequency-doubled laser pulses and transferability was demonstrated at the example of Pt-TiO₂ which also formed overgrown structures. The laser-treated Pt-CeO_x obtained a superior catalytic CO oxidation activity and superior cycle stability compared to the untreated sample. Since the activation energy of the catalyst didn't change on laser treatment the authors hypothesize that the overgrowth was only partial and active sites are still present. Overall the manuscript is mostly concise and informative addressing a novel method to induce/trigger SMSI in heterogeneous catalysts with reducible oxides in a controlled manner. A few questions are yet left open and need to be addressed to create a full picture before final publication can be recommended. The following points should be addressed:

1) Please comment on the role of chloride residues in laser-induced overgrowth. Provide evidence (e.g. EELS or XPS) that no chloride was present during laser treatment potentially affecting the overgrowth.

Response: Thank you for this comment. To explain the role of chloride residues in laser-induced overgrowth, the related experiments were performed, and the explanations have been revised in the manuscript:

(1) In this work, we try to minimize and avoid residual chloride ions in the preparation of fresh catalysts. After loading the precious metal on the oxide substrate, the sample was washed with a large amount of deionized water, until no obvious precipitation can be seen in wash water tested with AgNO_3 . The corresponding XPS results show that the general residual chloride content was less than 0.4 atom%, which was much lower than the added amount (about 8 atom%). It means that the residual chloride ions are minimal after extensive washing.

(2) Furthermore, from the current study, there is no direct evidence that residual chloride ions affect the formation of classic SMSI. Although the laser-induced construction of SMSI is different from the conventional H_2 reduction treatment, the commonalities from our current study far outweigh the differences. Therefore, we believe that the effect of residual chloride ions on laser-induced SMSI may be negligible.

Supplementary Fig. XPS full spectra of Pt/CeO₂

2) The authors convincingly discuss that minimization of the surface energy drives the overgrowth. With this interpretation, the minimization of surface energy is the driving force for the process. Yet, to fully understand the laser-induced processes the role of thermal excitation causing an improved atom mobility and laser-induced defect

formation (with defects altering the surface energy and hence the driving force) occur at the same time. Hence, the following important question remains: What drives the overgrowth, the laser-induced defect formation in CeO₂, or the thermal excitation? The question is the key to finalize the story and can be addressed with a simple reference experiment where both processes are disentangled:

Please provide a reference experiment where CeO₂ is irradiated without Pt NPs present and add the Pt-NPs after laser treatment. Please characterize the laser-treated cerium oxide by XPS and/or EELS and validate whether Ce³⁺ formed during laser treatment even though Pt-NPs are absent. This reference experiment thereby will also verify the discussed story on the field enhancement (absent in these experiments). Finalize the reference experiment by a respective HR-TEM investigation of the laser-treated CeO₂ after its functionalization with Pt-NPs to see if overgrowth did happen even without thermal excitation (only driven by the additional laser-induced Ce³⁺ defects) or not. Depending on the outcome the key role either of the laser-induced defects or the thermal excitation will be identified as a trigger for the overgrowth.

Response: Thank you for this comment. To fully understand the role of laser-induced defect formation and thermal excitation causing an improved atom mobility during the laser excitation. The related experiments were performed, and the explanations have been revised in the manuscript:

(1) A reference experiment where CeO₂ was irradiated without Pt NPs was performed. According to the XPS results (Supplementary Fig 11), there were no structural defects in the laser-treated CeO₂. It means that the same laser fluence irradiation without the enhanced electric field could not induce structural defects, and the localized electric field plays a vital role in the formation of structural defects.

(2) After that, Pt NPs were loaded on the laser-treated CeO₂ and subjected to heat treatment. According to the TEM images (Supplementary Fig 12), no overlayers were observed, suggesting that only thermal excitation could not induce SMSI. During the pulsed laser irradiation, the nanoconfined field essentially boosts the

formation of structural defects and atomic migration. On the other hand, thermal excitation could improve the atom mobility.

The sentence “To further explain the mechanism of laser-induced SMSI, a reference experiment where CeO₂ was irradiated without Pt NPs was performed. According to the XPS results (Supplementary Fig 11), there were no structural defects in the laser-treated CeO₂. It means that the same laser fluence irradiation without the enhanced electric field could not induce structural defects, and the localized electric field plays a vital role in the formation of structural defects. After that, Pt NPs were loaded on the laser-treated CeO₂ and subjected to heat treatment. According to the TEM images (Supplementary Fig 12), no overlayers were observed, suggesting that only thermal excitation could not induce SMSI. During the pulsed laser irradiation, the nanoconfined field essentially boosts the formation of structural defects and atomic migration. On the other hand, thermal excitation could improve the atom mobility.” was revised. (Page 11)

Supplementary Fig 11. XPS spectra of CeO₂ and laser-irradiated CeO₂

Supplementary Fig 12. TEM images of laser-treated CeO₂ loaded with Pt NPs

3) The interpretation of a partial overgrowth (brought forward due to the observed independence of the activation energy before and after laser treatment) is not concise with CO DRIFTS where the disappearance of the CO-Pt band was discussed as evidence for the overgrowth. If only partial overgrowth would be the case, a CO-Pt signal should have remained in DRIFTS but was not observed.

To remedy this issue; what is the active site for CO oxidation in the case of the Pt/CeO₂ catalyst? Please clearly state in the manuscript what the active sites are for this catalyst. Is it the Pt surface itself (as inferred, yet not directly mentioned by the partial overgrowth interpretation) or is the activity dominated by a Mars-Van-Krevelen mechanism at the interface of Pt and CeO₂ (also compare the observed activation enthalpy with literature values for pure Pt-NPs without CeO₂ to identify the role of CeO₂)? If a M-V-K based mechanism from an interaction between Pt-CeO₂ provides the active sites for CO oxidation the observed independence of the activation enthalpy and disappearance of the Pt-CO band would coincide. In this interpretation, the

overgrowth would mainly have increased the contact area between Pt and CeO₂ and hence the number of active sites while the overgrowth would not have changed the nature of the active sites (and hence no change in activation energy).

Please revise this aspect in your manuscript and clarify what the active sites are and why the Pt-CO signal disappears after laser treatment although the activation enthalpy remains constant. Partial overgrowth is insufficient to explain both latter observations.

Response: Thank you for this comment. Your suggestion is very enlightening for us to understand this experiment more deeply. It is generally accepted that CeO₂ loaded Pt NPs in CO catalytic oxidation follow Mars-Van-Krevelen mechanism and that the catalytic activity is size-dependent, where the perimeter of interface between the Pt NPs and CeO₂ is the active site of the reaction (Science, 2013, 341, 771. ACS Catal. 2015, 5, 5164). This mechanism also worked in our case, in which the improved catalytic activity of the laser-treated sample can be attributed to the increase in the number of Pt-CeO₂ effective interfacial sites. The apparent activation energy of the catalyst did not change significantly before and after laser irradiation, which were 59.34 kJ/mol and 58.99 kJ/mol, respectively (Activation energy is comparable to that reported in literature, Science, 2013, 341). The similarity of the apparent activation energy suggests that the catalysts before and after laser treatment follow similar reaction pathways. Therefore, the difference in activity probably mainly derived from the different number of effective active sites rather than rather than the exposed sites, which is in good agreement with previous studies (Science, 2013, 341, 771. ACS Catal. 2015, 5, 5164). It is already known from the substantial analysis above that the modification of CeO_x overlayer on the surface of Pt NPs, this will undoubtedly increment the interface between metal Pt and CeO_x. However, in general, the catalytic activity of the catalyst tends to degrade after the occurrence of the SMSI effect, due to the presence of overlayer on the metal surface. An important factor resulting in this catalyst passivation is that the overlayer is often dense and thus almost completely obscures the active sites of the catalyst. But in our case, it has been known from the HRTEM representation (Supplementary Fig. 6) that laser-induced formation

of CeOx overlayer is more distinctly different from the classical hydrogen reduction method. Due to the high-energy nature of the femtosecond laser, the overlayer induced with the laser is frequently discontinuous, and even a small number of Pt NPs are only partially encapsulated. The H₂-TPD spectra demonstrates the porous nature of the CeOx overlayer (Supplementary Fig. 24). When the femtosecond laser successfully constructs the SMSI effect between Pt and CeO₂, on the one hand, the discontinuous overlayers obscures the sites on the Pt surface, and on the other hand, it provides more active sites at the metal-support interface that are in direct contact with the reactants. This may be the main reason why the catalytic activity of Pt/CeO₂ increases rather than decays after the laser-induced SMSI effect.

Further minor comments:

4) Line 78-80: “However, the high temperature and high pressure caused by the nanosecond or longer pulse irradiation may induce unfavorable phase transformation with inferior properties²⁶.” Please specify. This sentence is too general. What phase transformations do you refer to?

Response: Thank you for this comment. The related descriptions were clarified and explained, and the explanations have been revised in the manuscript:

The description “However, the high temperature and high pressure caused by the nanosecond or longer pulse irradiation may induce unfavorable phase transformation with inferior properties.” was revised as “However, the high temperature and high pressure caused by the nanosecond or longer pulse irradiation may induce unfavorable phase transformation of anatase to rutile TiO₂ with inferior properties”. (Page 3)

5) Following sentence in line 80-83: “In addition, the noticeable photothermal effects involved in long-pulse laser irradiation might cause sintered NPs, resulting in decreased catalytic activity due to the sacrifice of active sites²⁷.” The sentence and the included statement is potentially misleading as the cited paper deals with laser melting where

aggregated NPs are molten together to form larger NPs. Non-aggregated particles will not show this sintering under ns-laser treatment but potential defect formation which is also referred to as laser defect engineering (LDL; pls see the current review of Amendola et al. DOI: 10.1002/chem.202000686).

Understandably, the author's intention in this paragraph was to draw attention to the importance of ultrafast lasers. The biggest advantage of ultrafast laser pulses is the strong electronic excitation (high electron temperature) and (compared to ns-lasers) lesser heat load in the lattice. Consequently, it was found that LDL of TiO₂ with picosecond laser pulses led to 2 times higher photocurrents discussed with the generation of defects when a sufficient number of laser pulses is applied (too high led to segregation and again lower activity). Nanosecond laser pulses in turn directly led to a gradual decrease of photocurrent potentially due to the formation of crystal defects in the lattice due to isochoric melting and subsequent rapid quenching (DOI: 10.1039/c5cp04296h). A more comprehensive review on this can also be found in the recent review of Amendola et al. (DOI: 10.1002/chem.202000686) The passage should be revised and clarified accordingly.

Response: Thank you for this comment. The related descriptions were clarified and explained, and the explanations have been revised in the manuscript:

The description “In addition, the noticeable photothermal effects involved in long-pulse laser irradiation might cause sintered NPs, resulting in decreased catalytic activity due to the sacrifice of active sites.” was revised as “For aggregated CuO NPs, the photothermal effects involved in long-pulse laser irradiation may cause sintering with the sacrifice of active sites. For non-aggregated TiO₂ NPs, when using nanosecond laser irradiation, a gradual decrease of photocurrent may result from the formation of bulk defects due to thermally initiated isochoric melting. When using a small number of picosecond pulses, the performances improved by a factor of two (ACS Applied Energy Materials 2018, 1, 5366, Chemistry 2020, 26, 9206).” (Page 3)

6) Figure 1a: the legend shows Ce, O, and C but no carbon can be seen in the figure. Also, the particle should be denoted as e.g. "Pt NP" for clarity

Response: Thank you for this comment. The legend in Fig. 1 has been revised according to the suggestion.

7) Figure 1b: The figures take some time to understand. To me, it seems more logical to switch the two figures to first show the 'zoomed-out' version and then the 'zoomed-in' version with the local field, etc. Same for Fig. 1c.

Response: Thank you for this comment. In order to better understand the images, Fig.1 has been revised according to the suggestion.

Fig. 1 Schematic of the ultrafast laser-induced SMSI in Pt/CeO₂ with highly catalytic stability. (a) Pt/CeO₂ nanostructure irradiated at ultrafast laser. (b) The structural defects and atomic migration induced by local field. (c) The highly catalytic stability obtained from the laser-induced SMSI.

8) Line 184-186: "It is well known that XPS is a surface analysis technique and that the Pt/Ce ratio on the surface decreases significantly with a CeO_x overlayer on the Pt surface, which was good agreement with XRD results." Where is this Pt/Ce ratio shown? Please provide how the at% of Ce and Pt (determined from XPS) changes before and after laser treatment.

Response: Thank you for this comment. The surface compositions determined from XPS were provided in the revised manuscript. After laser treatment, the Pt content decreased from 1.3 to 0.94 atom%, while the Ce content changed from 17.12 to 17.94 atom%. It means that the Pt/Ce ratio on the surface decreases with a CeO_x overlayer on the Pt surface, which was good agreement with XRD results.

The sentence “It is well known that XPS is a surface analysis technique. After laser treatment, the Pt content decreased from 1.3 to 0.94 atom %, while the Ce content changed from 17.12 to 17.94 atom %. It means that the Pt/Ce ratio on the surface decreases with a CeO_x overlayer on the Pt surface, which was good agreement with XRD results.” was revised. (Page 8)

9) Line 197: “...it is also clear that the Pt NPs were not oxidation after laser irradiation.”
Please check the language.

Response: Thank you for this comment. In order to avoid confusion, the related descriptions were clarified and explained. The description “it is also clear that the Pt NPs were not oxidation after laser irradiation.” was removed in the revised manuscript.

10) Line 202-204: “Generally, classic SMSI is reversible upon reversal treatment. Our analysis combining with the previous research reveal that the process of laser irradiation is somewhat reductive.” Recently, at the example of laser processing of Co₃O₄ and CoO, it was shown (DOI: 10.1002/cssc.201903186) that the initial oxidation state of the oxide affects whether the process is reductive or oxidative. Please consider including this aspect in the manuscript.

Response: Thank you for this comment. The related descriptions were clarified and explained, and the explanations have been revised in the manuscript:

The description “Our analysis combining with the previous research reveal that the process of laser irradiation is somewhat reductive.” was revised as “During the pulsed laser processing, the oxide could be reduced or oxidized, which

depended on the initial oxidation state (ChemSusChem 2020, 13, 520). Under the experimental condition, the CeO₂ support was reduced with the increase in Ce³⁺ concentration.” (Page 9)

11) Line 208-211: “This result suggests that laser-induced construction of SMSI should be irreversible in the oxidizing atmosphere, which was not consistent with classic SMSI and the reasons for this still need to be further studied.” Please comment if there is any evidence of Pt-Ce-mixed oxides formed in the overgrown layer.

Response: Thank you for this comment. The related descriptions were clarified and explained, and the explanations have been revised in the manuscript:

(1) We have modified this part of the in-situ CO-DRIFT analysis in the manuscript accordingly. More in-depth studies have shown that laser-induced SMSI is reversible in oxidizing atmospheres, which is similar to conventional H₂ reduction. In the original version of the manuscript, the reason why we thought that laser-induced SMSI was irreversible in an oxidizing atmosphere was that the SMSI effect was observed to be still present after reoxidation of laser-Pt/CeO₂ at 500°C (TEM images and in-situ CO-DRIFT confirm this). However, recent experiments have found that the SMSI effect of laser-Pt/CeO₂ disappears when the oxidation temperature is greater than 500°C or extending the oxidation time at 500°C. In this system of Pt/CeO₂, there is still a significant difference between the laser-induced construction of SMSI and the conventional SMSI constructed by H₂ reduction. The overcoat formed by laser-induced SMSI is not dense but tends to be thicker and more crystalline, thus providing a better resistance to oxidation. If we had been able to carefully compare the TEM images of laser-Pt/CeO₂ before and after oxidation at 500°C it would have been possible to find this (The overlayer becomes thinner after oxidation at 500°C for 1 h). In conclusion, the laser-induced SMSI effect is reversible under oxidizing atmosphere.

(2) Previous studies have shown that alloying phenomena of Pt/CeO₂ catalysts can be detected only at the highest reduction temperature of 1073-1223 K, which indicates that the conditions for CePt alloys are extremely severe. (Journal of

catalysis 169, 510, 1997; Catalysis Letters 76, 3, 2001). In our case, no alloying was found from the HRTEM observations. Even for particles with thicker CeO_x overlays, no alloying was observed at the interface between the CeO_x layer and the Pt NPs (Supplementary Fig 5).

Supplementary Fig 5. HRTEM images of laser-Pt/CeO₂

12) In a previous paper (DOI: 10.1039/c5cp04296h) the inclusion of AuNP in ZnO with an increasing number of laser cycles (= number of laser pulses) well in line with the discussion on irradiation duration and increasing thickness of the overgrown layer and hence could be mentioned in this context.

Response: Thank you for your careful review of our manuscript. Your constructive comments and suggestions have helped us to further improve our manuscript. According to the comment, the description “This finding was similar to observations in the pulsed laser processing of Au/ZnO. When increasing the number of laser pulses, Au NPs were totally included into ZnO support (Physical chemistry chemical physics 2015, 17, 29311)” was revised. (Page 11)

Reviewer #2 (Remarks to the Author):

The authors report the use of a femtosecond laser to induce strong metal support interactions (SMSI) in a Pt/CeO₂ material (i.e., overcoating of Pt by the suboxide of ceria). It is argued that such SMSI are beneficial for low-temperature oxidation of CO relative to the untreated Pt/CeO₂ that does not contain such SMSI. The manuscript compares characterization data and catalytic activity data for laser-treated and untreated Pt/CeO₂. To understand the relevance of this exotic way to induce SMSI (with a femtosecond laser), it is necessary to provide the respective benchmark data for the Pt/CeO₂ material with conventionally-induced SMSI, i.e. by pretreating Pt/CeO₂ under H₂ prior to the catalytic test (and doing selected characterization). Without this data, the Manuscript appears incomplete and assessment of its importance and impact is difficult. That being said, it well may be that the overcoatings that has been obtained in an aqueous suspension without removal of air using laser (oxidizing environment) and under H₂ flow (reducing environment) are different chemically. Yet it needs to be demonstrated in the first place.

Response: Thank you for your careful review of our manuscript. Your constructive comments and suggestions have helped us to further improve our manuscript. As mentioned in the manuscript, although the study of constructing SMSI by hydrogen reduction has been developed for decades, new methods for constructing SMSI (reductive and non-reductive support) under ambient conditions are necessary. One of the biggest bottlenecks is that catalyst systems with SMSI effect are mostly limited to reducible metal oxide loaded platinum group metals. Even classical catalyst systems like Au/TiO₂ were not found to have SMSI effects until 2017 (Sci. Adv 2017, 3, 1700231).

Therefore, it is valuable to find a methodology that can universally achieve SMSI effects on both reductive and non-reductive support. This is the original purpose of our experiment on laser-induced SMSI effect. It is well known that the prerequisite for the construction of SMSI effects is the ability to achieve support activation to form migratory metastable surface species. The study found that laser irradiation can activate the surface of materials to produce unstable species

(Chemical reviews 2017,117, 3990). When the high-energy laser interacts with the particles dispersed in the solution, the surface structure is reconfigured or even melted, and the melting is simultaneously quenched, this process that leaves large number of defects on the surface of the material, leaving the surface species in a migratory sub-stable state, thus achieving activation of the material surface. The above phenomenon provides the preliminary conditions for the successful construction of the SMSI effect. Since our group had previously studied CeO₂-based materials and applied it to catalytic reactions, the classical system of Pt/CeO₂ was chosen as the first object of investigation. However, the experiment took many detours in the absence of experience, so the SMSI effect was only observed in two common systems, Pt/CeO₂ and Pt/TiO₂, constructed with laser induction even before the submission of this manuscript. But thankfully, in recent work, we have also observed classical SMSI effects in Au/TiO₂, Au/MgO, Pt/Al₂O₃, Pd/TiO₂, Pt/SiO₂ systems by tuning the laser process and other experimental parameters. Although there are still many problems, at the very least, it is possible to demonstrate the potential of laser irradiation to universally construct SMSI effects in these classical catalyst systems (reducible and unreducible support; Pt group metal and Au, etc.). I think it may be useful to broaden the boundary of SMSI effect and improve the understanding of non-homogeneous catalysis.

To further demonstrate the structural specificity of the laser-induced SMSI effect, the classical method of hydrogen reduction was also adopted to construct SMSI on Pt/CeO₂. According to a previous report, fresh-Pt/CeO₂ was calcined at 700°C under hydrogen atmosphere for 2h (Journal of catalysis 169, 510, 1997). The HRTEM images revealed that an amorphous and dense CeO_x overlayer was formed on the surface of Pt NPs after H₂ reduction, which is more significantly different from the non-dense overlayer formed by laser-induced SMSI, but is consistent with previous reports (Supplementary Fig. 25) (Catalysis Letters 76, 3, 2001). The in-situ CO-DRIFT shown that the adsorption of CO is suppressed after high temperature reduction of Pt/CeO₂ by hydrogen (Supplementary Fig. 26). The above results all demonstrate the formation of SMSI between Pt and CeO₂ after

high temperature reduction. CO oxidation was also used as a probe reaction to study the changes in its activity before and after the formation of SMSI. It was found that the catalytic activity of Pt/CeO₂ showed a significant decay after the formation of SMSI under hydrogen reduction, and its complete conversion temperature increased from 170°C to 230°C for fresh samples. However, the catalytic activity was restored after reoxidation of the H₂-Pt/CeO₂ sample under air atmosphere at 600°C, and its complete conversion of CO was achieved at 160°C (Supplementary Fig. 27). The above experimental phenomena are in good agreement with the classical SMSI, but differ significantly from the laser-induced SMSI effect. The root cause of this difference is perhaps the difference in the structure of the CeO_x overlayer, as mentioned above. Although the H₂ reduction construct SMSI effect is also able to increase the interface between the Pt and CeO_x, but the dense CeO_x overlayer formed on the surface of Pt NPs makes it difficult for the reactants to directly contact the interfacial sites, which instead reduces the activity of the catalyst.

Figure S25. TEM images of (a) fresh-Pt/CeO₂; (b-c) Pt/CeO₂- H₂700 °C; (d) Pt/CeO₂-

H₂700 °C -O₂600 °C

Supplementary Fig 26. In situ CO-DRIFT of fresh Pt/CeO₂ and H₂-Pt/CeO₂

Figure S27. CO oxidation curves of fresh-Pt/CeO₂, H₂-Pt/CeO₂ and H₂-Pt/CeO₂-O₂ catalyst.

Additional questions (according to the flow of the Manuscript but not their importance) are as follows:

1) Why is it relevant to include “ultrafast” in the title “Ultrafast laser-induced strong metal-support interactions in Pt/CeO₂ with highly catalytic stability”? The recipe to

prepare materials takes up to 1 hour, this is hardly ultrafast. “Highly catalytic stability” is confusing.

Response: Thank you for this comment. The related descriptions were clarified and explained, and the explanations have been revised in the manuscript:

(1) The processing time (1h) was not ultrafast. This word was not used to describe the laser. Femtosecond laser is a kind of ultrafast laser which has the characteristic of ultrashort pulse duration (less than few picoseconds).

(2) According to the comment, the description “Ultrafast laser induced strong metal-support interactions in Pt/CeO₂ with highly catalytic stability” was revised as “Ultrafast laser induced strong metal-support interactions in Pt/CeO₂”. (Page 1)

2) The authors refer to ZnO as a non-reducible oxide. This is not correct, ZnO is regarded as reducible (in contrast to silica, alumina, magnesia, etc).

Response: Thank you for this comment. The related descriptions were clarified and explained, and the explanations have been revised in the manuscript:

The description was revised as “In order to break the bottleneck, classical SMSI was achieved between Au and non-reducible MgO through CO₂-induced activation of the oxide surface (Nature Catalysis 2021, 4, 418)”. (Page 2)

3) Simple H₂ pretreatment typically used to induce SMSI is presented as requiring “harsh experimental conditions (high temperature, explosive reductant (H₂))” – this is an overstatement, especially relative to the complexity of the method presented in this work.

Response: Thank you for this comment. The related descriptions were clarified and explained, and the explanations have been revised in the manuscript:

The description “These methods are limited by harsh experimental conditions (high temperature, explosive reductant (H₂)) and complex procedures” was revised as “Thermally induced reactions in specific gaseous atmospheres are generally required in traditional procedures. Therefore, new methods for

constructing SMSI under ambient conditions are still essential to design high-performance catalysts and understand SMSI effects in more depth”. (Page 3)

4) The authors write that they have developed a material “which possesses a strong quenching effect to generate metastable nanostructures with extraordinary properties.” – it is not clear what extraordinary properties have been discovered in the present work. None of the properties discussed appears extraordinary.

Response: Thank you for this comment. This sentence was intended to illustrate the advantages of pulsed lasers (Science 2017, 358, 640; Nature Catalysis 2019, 2, 1107). According to the comment, the description “which possesses a strong quenching effect to generate metastable nanostructures with extraordinary properties” was revised as “which possesses a strong quenching effect to generate metastable nanostructures”. (Page 3)

5) Figures 1a and 1b contain a label of carbon, however the structures presented are carbon-free.

Response: Thank you for this comment. The legend in Fig. 1 has been revised according to the suggestion.

6) The intensity of diffraction peaks has decreased after the laser treatment. The authors propose that “The weakening of the diffraction peaks can therefore be explained by the presence of the CeO_x overlayer.” This explanation does not look convincing, given the small thickness of the overlayer observed by TEM imaging.

Response: Thank you for this comment. The related descriptions were clarified and explained, and the explanations have been revised in the manuscript:

Under ultrafast laser irradiation, abundant metastable structures could be formed in metal NPs because of the strong quenching effect (Small 2020, 16, 2001101). The weakening of the diffraction peaks can be explained by the surface-induced poor crystallinity (Nature Communications 2020, 11, 3220).

The sentence “Under ultrafast laser irradiation, abundant metastable structures could be formed in metal NPs because of the strong quenching effect. The weakening of the diffraction peaks can be explained by the surface-induced poor crystallinity” was revised. (Page 7)

7) The XPS peak at ca. 72.05 eV in Pt/CeO₂ is attributed to Pt^{δ+}. The intensity of this peak is found to increase after laser irradiation, explained by the formation of Pt-O at the interface between Pt and CeO_x overlayer. Next, “transferring electrons from Pt to CeO_x” is suggested. It has been reported in the literature that the SMSI effect (geometric effect, overcoating) often occurs along with the EMSI effect, i.e. electronic metal support interaction. The latter includes the electron transfer, as mentioned, yet the transfer of electrons in the opposite direction is typical, that is from the reduced CeO_x to a supported noble metal such as Pt, Ru, etc. This creates M(δ⁻) states, which are more reduced in XPS than M(0) states. Based on this, the suggested “transferring electrons from Pt to CeO_x” may not be correct and requires additional experimental evidence. Can it be that the laser exposure formed small Pt clusters or atoms?

Response: Thank you for this comment. The related descriptions were clarified and explained, and the explanations have been revised in the manuscript:

(1) Smaller Pt NPs may indeed be formed under laser irradiation. However, the size of the NPs did not decrease significantly after laser irradiation. On the contrary, a small amount of Pt particles was found to grow after laser irradiation. Therefore, we speculate that the Pt nanoclusters produced by laser irradiation may not be the main reason for the increase in intensity of the Pt^{δ+} characteristic peak.

(2) As mentioned, Pt/CeO₂ is first dispersed in water and then laser irradiated, and the whole process is exposed to air. On the one hand, water has a certain solubility of oxygen, like the solubility of oxygen at 2°C is 9.17 mg/L. On the other hand, laser irradiation is performed with continuous agitation, which means that the catalysts are all directly exposed to air. In the presence of oxygen, the local heat generated by laser irradiation may induce Pt NPs to bind to oxygen.

Moreover, previous studies have shown that in the presence of oxygen involved in the construction of SMSI, Pt-O bonds are generated at the interface between Pt NPs and the overlayer (Nature communications 2020, 11, 1-8). In summary, we speculate that the involvement of oxygen is the main reason for the increase in intensity of the Pt^{δ+} characteristic peak after laser irradiation.

8) The plot with Pt/Ce ratios is missing. This data is essential to demonstrate the occurrence of SMSI (in addition to the TEM data, which is a local method and probes only a fraction of specimen). It is suggested that the authors add this plot for materials of Figure 4, i.e., after 20, 40 and 60 min of laser irradiation. Is it possible to present and discuss Pt/Ce(4+) and Pt/Ce(3+) ratios for these samples? How would those ratios compare to the reference Pt/CeO₂-H₂ material?

Response: Thank you for this comment. To further demonstrate the occurrence of SMSI, the related experiments were performed, and the explanations have been revised in the manuscript:

The description “According to the XPS results (Supplementary Fig. 9), the Pt/Ce ratio on the surface of specimens were 0.076 (Fresh-Pt/CeO₂), 0.0459 (Pt/CeO₂-laser-20 min), 0.0394 (Pt/CeO₂-laser-40 min), 0.0344 (Pt/CeO₂-laser-60 min), and 0.0557 (Pt/CeO₂-H₂), respectively. It means that Pt NPs were totally encapsulated with the increased exposure time, and the thickness of the overlayer was higher, indicating the enhanced structural reorganization (Fig. 4c, d).” was revised. (Page 11)

Supplementary Fig 9. XPS results of the Pt/Ce ratio on the surface of specimens (Fresh-Pt/CeO₂, Pt/CeO₂-laser-20 min, Pt/CeO₂-laser-40 min, Pt/CeO₂-laser-60 min, and Pt/CeO₂-H₂).

9) IR data with CO probe molecule in Fig. 3f is not suitable because too large amounts of CO have been admitted to the cell, which creates large peaks due to CO gas-phase absorption. This covers area of more blue-shifted CO peaks (up to ca. 2175 cm⁻¹). So it is not possible to evaluate the presence or absence of Pt(δ+)-CO adducts. Rather use pulses of small amounts of CO to the outgassed specimen, from ca. 0.1 mbar to ca. 1 mbar.

Response: Thank you for this comment. After the CO adsorption saturation, the gaseous CO was flashed off by evacuation and then the spectra were collected as follows. With this method, the large peaks of gaseous CO were successfully eliminated.

Fig. 3f In situ CO-DRIFT of fresh Pt/CeO₂ and laser-irradiated Pt/CeO₂

10) How similar (or dissimilar) are carbonate bands on discussed materials? It can be that carbonates form after oxidation of CO by the lattice oxygen already at room temperature.

Response: Thank you for this comment. The peaks in the carbonate band have almost disappeared after laser irradiation. This is probably due to the abstraction of lattice oxygen from the CeO₂ surface after irradiation, which makes it difficult for CO to be oxidized under these conditions and thus no carbonate adsorption band. Because of this, when laser-Pt/CeO₂ was oxidized at high temperature, the carbonate adsorption band appeared again, but the intensity was significantly weakened.

11) Did the authors attempt to evaluate the metal surface area in their materials? H₂-TPD can be a method of choice. Having this information will allow to evaluate the suggested porosity of the overlayer and availability of the metal surface. It is advised to provide this data. Likewise, CO-TPD measurement can also be deployed. The porosity of CeO_x is suggested but no experimental data directly supporting this speculation has been provided.

Response: Thank you for this comment. The H₂-TPD spectra was shown below. The catalyst exhibited three H₂ desorption regions, the desorption in the low temperature region (I) can be attributed to H species physically adsorbed on the catalyst surface; the medium temperature region (II) can belong to H₂ strongly chemisorbed on the Pt surface, while the high temperature region (III) may be ascribed to hydrogen spillover (from the support to the Pt) or H species desorbed from the subsurface of the supports (J Catal. 1999, 186, 279. J Catal. 1992, 137, 1. J Nanopart Res. 2016, 18, 66). After laser irradiation, the intensity of the H₂ desorption peak corresponding to laser-Pt/CeO₂ was significantly weakened, but obvious desorption peaks could still be observed in the low and medium temperature desorption regions, which indicates that the CeO_x overlayer possesses porous features, which was consistent with the TEM and CO-DRIFT results.

Supplementary Fig 24. H₂-TPD spectra of fresh Pt/CeO₂ and laser-irradiated Pt/CeO₂

12) Figure 4a and respective discussion in the text. Authors are asked to justify (with citation to relevant literature) their assertion that oxygen vacancy site in ceria is necessarily reducing two Ce⁴⁺ ions to two Ce³⁺ ions. Is it correct that this electron density is localized fully at the oxygen vacancy site (and not delocalized), disregard of

the density of oxygen vacancy sites? EPR data shows only a minor signal increase in the laser-treated sample.

Response: Thank you for this comment. The related descriptions were clarified and explained, and the explanations have been revised in the manuscript:

The description “On the surface of CeO₂, O atoms donated the electrons to the Ce atoms. Therefore, Ce⁴⁺ accepted the electron to form the Ce³⁺, and the O atoms could be peeled off from the surface of the CeO₂ to form the oxygen vacancies (ACS Appl. Mater. Interfaces 2015, 7, 16070). In the crystal structure, two of the cerium ions are replaced by trivalent ions, between which an oxygen vacancy appears (Materials Today 2014, 17, 349).” was revised. (Page 10)

13) Cycles of CO oxidation are discussed, but what is a cycle? This has not been defined in the Manuscript or Supplementary Information.

Response: Thank you for this comment. The related descriptions were clarified and explained, and the explanations have been revised in the manuscript:

The description “It means the conversion of CO from 90 to 200 °C on laser-Pt/CeO₂ catalyst.” was revised. (Page 17)

14) Arrhenius plots in Fig. 6b show the difference in E_a for the two catalysts studied of merely 0.3 kJ mol⁻¹. Yet the temperature difference to reach 50% CO conversion with these catalysts is ca. 50 °C. Does this make sense?

Response: Thank you for this comment. The related descriptions were clarified and explained, and the explanations have been revised in the manuscript:

The similarity of the apparent activation energy suggests that the catalysts before and after laser treatment follow similar reaction pathways. Therefore, it can be considered the difference in activity probably mainly derived from the different number of effective active sites rather than the exposed sites, which is in good agreement with previous studies (Science 2013, 341, 771; ACS Catal 2015, 5, 5164). It is generally accepted that CeO₂-loaded Pt NPs in CO catalytic oxidation follow Mars-Van-Krevelen mechanism and that the catalytic activity is size-dependent,

where the perimeter of interface between the Pt NPs and CeO₂ is the active site of the reaction. This mechanism also worked in our case, in which the improved catalytic activity of the laser-treated sample can be attributed to the increase in the number of Pt-CeO₂ effective interfacial sites due to the discontinuous CeO_x overlayer. In conclusion, similar apparent activation energies are relevant to our deeper understanding of laser-induced SMSI.

15) Figure S2, y-axis. Replace “probability” by “particle count”. Without this information it is not clear if 10 particles have been analyzed or 200 particles.

Response: Thank you for this comment. In order to better understand the images, Supplementary Fig. 2 has been revised according to the suggestion.

Supplementary Fig. 2. The size distributions of Pt NPs (a) before and (b) after laser irradiation deduced from statistics of TEM images.

16) Reporting BET surface area with the precision to the third decimal point (Figure S4) is unreasonable. The error of the measurement is larger than what the authors suggest with their third decimal point accuracy.

Response: Thank you for this comment. The BET surface area in Supplementary Fig. 4 has been revised according to the suggestion.

The description “But the BET surface area of the fresh-Pt/CeO₂ and laser-irradiated Pt/CeO₂ was about 52.7 m²/g and 46.1 m²/g, respectively (Supplementary Fig. 4a)” was revised. (Page 7)

17) Can the authors demonstrate SMSI effect for Pt particles on more challenging irreducible supports? Ceria and titania are well known for SMSI effects, this is easily achieved by a simple H₂ pretreatment, therefore current results are not very surprising.

Response: Thank you for your careful review of our manuscript. Your constructive comments and suggestions have helped us to further improve our manuscript: According to the comment, we used this strategy to fabricate Pt/Al₂O₃, Pt/SiO₂, and Au/MgO NPs, and the laser-induced SMSI was also obtained, showing its possibility of extending to more challenging irreducible supports.

The sentence “In addition, the laser-induced SMSI was also obtained in Pt/TiO₂, Pd/TiO₂, Au/TiO₂, Pt/Al₂O₃, Pt/SiO₂, and Au/MgO NPs, suggesting that this strategy can be extended to other metal/metal oxide materials (Supplementary Fig. 17-22)” was revised. (Page 14)

Supplementary Fig 17. (a-c) TEM images of fresh-Pt/TiO₂. (d-f) TEM images of laser-irradiated Pt/TiO₂.

Supplementary Fig 18. (a-b) TEM images of fresh- Pd/TiO₂. (c-f) TEM images of laser-irradiated Pd/TiO₂.

Supplementary Fig 19. (a-c) TEM images of fresh- Au/TiO₂. (d-f) TEM images of laser-irradiated Au/TiO₂.

Supplementary Fig 20. (a-c) TEM images of fresh- Pt/Al₂O₃. (d-f) TEM images of laser-irradiated Pt/Al₂O₃.

Supplementary Fig 21. (a-c) TEM images of fresh- Pt/SiO₂. (d-f) TEM images of laser-irradiated Pt/SiO₂.

Supplementary Fig 22. (a-c) TEM images of fresh- Au/MgO. (d-f) TEM images of laser-irradiated Au/MgO.

Supplementary Table 1. Comparison of the SMSI construction between the previous procedures and the pathway used in this work.

	Material	Support properties	Method	Reference
1	Au/TiO ₂	Reductive	H ₂ treatment	Sci. Adv. 2017, 3, 1700231; Nat. Commun. 2020, 11, 5811
2	Au/TiO ₂	Reductive	Melamine treatment	Nat. Commun. 2019, 10, 5790
3	Au/TiO ₂	Reductive	Wet-chemistry	J. Am. Chem. Soc. 2019, 141, 2975
4	Au/TiO ₂	Reductive	Sacrificial coating	J. Am. Chem. Soc. 2016, 138, 16130.
5	Au/TiO ₂	Reductive	Thermal annealing	Angew. Chem. Int. Ed. 2017, 56, 4494
6	Pt/TiO ₂	Reductive	H ₂ treatment	Nat. Commun. 2020, 11, 3220
7	Au/MgO	Irreductive	CO ₂ treatment	Nat. Catal. 2021, 4, 418
8	Au/SiO ₂	Irreductive	Deposition-precipitation	Nat. Commun. 2020, 11, 558
9	Au/ hydroxyapatite	Irreductive	H ₂ treatment	J. Am. Chem. Soc. 2016, 138, 56
10	Pt/CeO ₂ , Pt/TiO ₂ , Pd/TiO ₂ , Au/TiO ₂ , Pt/Al ₂ O ₃ , Pt/SiO ₂ , and Au/MgO	Reductive /irreductive	Ultrafast laser treatment	This work

REVIEWER COMMENTS

Reviewer #1 (Remarks to the Author):

The manuscript has been sufficiently revised and all open questions were addressed accordingly. The manuscript describes a very elegant and interesting study that provides very valuable insights. I recommend accepting the manuscript in its now revised state.

Reviewer #2 (Remarks to the Author):

The revised manuscript has improved. In particular the added examples of SMSI with Pt or Au on irreducible supports (alumina, silica, MgO) and benchmarking data with H₂-induced SMSI on Pt/CeO₂ has strengthened the work. This helps, from the results standpoint, to make a case for Nat. Commun.

The scientific prose in the Manuscript is often unsatisfactory, both in terms of the level of discussion, or unclear statements, which are often too broad and imprecise (as has also been pointed out by Reviewer 1). In what follows, several examples are provided (a critical read of the text will identify more of such cases). Comments below follow in the order of appearance in the text.

- 1) Title. "Ultrafast laser induced strong metal-support interactions" is unclear because it may be understood as metal-support interactions are ultrafast. Change for "Strong metal-support interactions induced by an ultrafast laser" or a similar title.
- 2) "Strong metal-support interactions (SMSI) are an important concept in heterogeneous catalysis" and "we propose a new concept of ultrafast laser-induced SMSI". SMSI is not a concept but an experimentally observed phenomenon.
- 3) "This special strategy" – delete "special"
- 4) "Although the mechanisms of the SMSI effect remain controversial to date" – what is controversial about the mechanisms of SMSI? Specify or delete.
- 5) "When the high-energy laser interacts with the NPs dispersed in the solution, the surface structure is reconfigured. A large number of defects formed on the surface of the material, leaving the surface species in a migratory sub-stable state" – this reads as if supported NPs first leach into solution, where the interaction with the high-energy laser irradiation takes place. Is this what authors want to say? What is meant by reconfiguration of the surface structure of NPs (and not the support)? What is meant by "the material"? The first sentence discusses NPs, however, here the support was likely meant, in place of the NPs. What is meant by the surface defects? If these are oxygen vacancy sites, then say it explicitly, as there exist other surface defects in addition to Ov sites. Can authors define "migratory sub-stable state"? Is it a suboxide? I take those two cited sentences and questions they rise as an example of how unclear and unspecific the prose of this Manuscript could be.
- 6) "The mechanism underlying the SMSI formation was revealed" – while there are experiments towards understanding of the mechanism (images in Fig. 4, comparison of experiments with and without Pt particles on CeO₂), I still don't quite follow the explanation of the physics behind the developed process.

Is this a thermal effect owing to the heating localized at the interface between the NP and the support, as cartoons seem to suggest (Fig. 2a; it is known that upon heating CeO₂ will lose lattice oxygen and form CeO_{2-x})? How is the calculated electric field distribution (Fig. 2b-d) relates to the temperature distribution at the NP and at the 3-phase boundary (between NP, support and reaction medium)? How important is involvement of localized plasmon resonance at the Pt/CeO₂ interface to induce the SMSI effect? Or is the plasmonic optical response too weak due to the small size of NPs? The heat accumulation and dissipation at the interface probably depends on the interaction between charge carriers and phonons.

7) “The highly catalytic stability” – the high catalytic stability

8) “In our case, no alloying [between Ce and Pt] was found from the HRTEM observations.” – it can be difficult to find evidence of alloying by HRTEM if the relative fraction of Ce is low. Maybe an additional argument can be made based on the XPS data?

9) “According to the XPS results (Supplementary Fig. 9), the Pt/Ce ratio on the surface of specimens were 0.076 (Fresh-Pt/CeO₂), 0.0459 (Pt/CeO₂-laser-20 min), 0.0394 (Pt/CeO₂-laser-40 min), 0.0344 (Pt/CeO₂-laser-60 min), and 0.0557 (Pt/CeO₂-H₂), respectively. It means that Pt NPs were totally encapsulated with the increased exposure time, and the thickness of the overlayer was higher, indicating the enhanced structural reorganization” – define at what Pt/Ce ratio the encapsulation becomes “total”, and how this correlates with the argument that, according to the catalytic data, the overlayer is porous. In addition, I do not understand what “enhanced structural reorganization” means.

10) “After that, Pt NPs were loaded on the laser-treated CeO₂ and subjected to heat treatment.” – define the treatment.

Response to reviewers

Reviewer #1 (Remarks to the Author):

The manuscript has been sufficiently revised and all open questions were addressed accordingly. The manuscript describes a very elegant and interesting study that provides very valuable insights. I recommend accepting the manuscript in its now revised state.

Response: Thank you very much for your careful review and high evaluation of our manuscript. Your constructive comments and suggestions have helped us to further improve our manuscript.

Reviewer #2 (Remarks to the Author):

The revised manuscript has improved. In particular the added examples of SMSI with Pt or Au on irreducible supports (alumina, silica, MgO) and benchmarking data with H₂-induced SMSI on Pt/CeO₂ has strengthened the work. This helps, from the results standpoint, to make a case for Nat. Commun.

The scientific prose in the Manuscript is often unsatisfactory, both in terms of the level of discussion, or unclear statements, which are often too broad and imprecise (as has also been pointed out by Reviewer 1). In what follows, several examples are provided (a critical read of the text will identify more of such cases). Comments below follow in the order of appearance in the text.

Response: Thank you very much for your careful review of our manuscript. Your constructive comments and suggestions have helped us to further improve our manuscript. In order to better present this work, the related discussion and statements have been revised in the manuscript.

1) Title. “Ultrafast laser induced strong metal-support interactions” is unclear because it may be understood as metal-support interactions are ultrafast. Change for “Strong metal-support interactions induced by an ultrafast laser” or a similar title.

Response: Thank you for this comment. In order to avoid confusion, the title “Ultrafast laser induced strong metal-support interactions” was revised as “Strong metal-support interactions induced by an ultrafast laser”.

2) “ Strong metal-support interactions (SMSI) are an important concept in heterogeneous catalysis” and “we propose a new concept of ultrafast laser-induced SMSI”. SMSI is not a concept but an experimentally observed phenomenon.

Response: Thank you for this comment. The related descriptions were clarified and explained, and the explanations have been revised in the manuscript:

The description “Strong metal-support interactions (SMSI) are an important concept in heterogeneous catalysis. Constructing SMSI is an effective means of regulating the interfacial properties of noble metal-based supported catalysts” was revised as “Supported metal catalysts play a crucial role in the modern industry. Constructing strong metal-support interactions (SMSI) is an effective means of regulating the interfacial properties of noble metal-based supported catalysts”. (Page 1)

The description “we propose a new concept of ultrafast laser-induced SMSI” was revised as “we propose a new strategy of ultrafast laser-induced SMSI”. (Page 1)

3) “This special strategy” – delete “special”

Response: Thank you for this comment. The related descriptions were clarified and explained, and the explanations have been revised in the manuscript:

The description “This special strategy of constructing SMSI” was revised as “This strategy of constructing SMSI”. (Page 1)

4) “ Although the mechanisms of the SMSI effect remain controversial to date” – what is controversial about the mechanisms of SMSI? Specify or delete.

Response: Thank you for this comment. The related descriptions were clarified and explained, and the explanations have been revised in the manuscript:

The description “Although the mechanisms of the SMSI effect remain controversial to date” was deleted according to the suggestion. (Page 2)

5) “When the high-energy laser interacts with the NPs dispersed in the solution, the surface structure is reconfigured. A large number of defects formed on the surface of the material, leaving the surface species in a migratory sub-stable state” – this reads as if supported NPs first leach into solution, where the interaction with the high-energy laser irradiation takes place. Is this what authors want to say? What is meant by reconfiguration of the surface structure of NPs (and not the support)? What is meant by “the material”? The first sentence discusses NPs, however, here the support was likely meant, in place of the NPs. What is meant by the surface defects? If these are oxygen vacancy sites, then say it explicitly, as there exist other surface defects in addition to Ov sites. Can authors define “migratory sub-stable state”? Is it a suboxide? I take those two cited sentences and questions they rise as an example of how unclear and unspecific the prose of this Manuscript could be.

Response: Thank you for this comment. The related descriptions were clarified and explained, and the explanations have been revised in the manuscript:

The description “When the high-energy laser interacts with the NPs dispersed in the solution, the surface structure is reconfigured. A large number of defects formed on the surface of the material, leaving the surface species in a migratory sub-stable state, thus achieving activation of the material surface” was revised as “When the high-energy laser interacts with the metal oxides dispersed in the solution, the surface structure is reconfigured. Oxygen vacancies formed on the surface of the metal oxides, leaving the surface species in a suboxide state, thus achieving activation of the metal oxides surface”. (Page 3)

6) “The mechanism underlying the SMSI formation was revealed” – while there are experiments towards understanding of the mechanism (images in Fig. 4, comparison of experiments with and without Pt particles on CeO₂), I still don’t quite follow the explanation of the physics behind the developed process. Is this a thermal effect owing

to the heating localized at the interface between the NP and the support, as cartoons seem to suggest (Fig. 2a; it is known that upon heating CeO_2 will lose lattice oxygen and form CeO_{2-x})? How is the calculated electric field distribution (Fig. 2b-d) relates to the temperature distribution at the NP and at the 3-phase boundary (between NP, support and reaction medium)? How important is involvement of localized plasmon resonance at the Pt/ CeO_2 interface to induce the SMSI effect? Or is the plasmonic optical response too weak due to the small size of NPs? The heat accumulation and dissipation at the interface probably depends on the interaction between charge carriers and phonons.

Response: Thank you for this comment. In order to better understand the mechanism underlying the SMSI formation, the related descriptions were clarified and explained, and the explanations have been revised in the manuscript:

(1) When the ultrafast laser irradiated on the Pt/ CeO_2 NPs, the electric field would be confined in the localized Pt/ CeO_2 interface and enhanced, which arises from the localized plasmon resonance. The enhanced field can induce the nonlinear effects and ionize CeO_2 . Specifically, when a flux of photons (1.55 eV at 800 nm) was injected, the bounded electrons of CeO_2 were excited to the conduction band by multiphoton absorption, leaving the holes in the valence band. On the surface of CeO_2 , O atoms donated the electrons to the Ce atoms. Therefore, Ce^{4+} accepted the electron to form the Ce^{3+} , and the O atoms could be peeled off from the surface of the CeO_2 to form the oxygen vacancies. Ce atoms may migrate randomly with associated oxygen atoms on the unstable surface to form reorganized structures at the perimeter interface, and Pt NPs were encapsulated.

(2) To further explain the role of nanoconfined electric field, a reference experiment where CeO_2 was irradiated without Pt NPs was performed. It was found that the same laser fluence irradiation without the enhanced electric field could not induce surface defects. After that, Pt NPs were loaded on the laser-treated CeO_2 and subjected to heat treatment (under argon atmosphere at 500°C for 2 h). No similar overlayers were observed, suggesting that only thermal excitation could not induce SMSI. During the pulsed laser irradiation, the

nanoconfined field essentially boosts the formation of Ce³⁺/oxygen vacancy and metastable CeO_x migration. On the other hand, thermal excitation could improve the metastable CeO_x mobility.

7) “The highly catalytic stability” – the high catalytic stability

Response: Thank you for this comment. The related descriptions were clarified and explained, and the explanations have been revised in the manuscript:

The description “The highly catalytic stability” was revised as “The high catalytic stability”. (Page 4)

8) “In our case, no alloying between Ce and Pt was found from the HRTEM observations.” – it can be difficult to find evidence of alloying by HRTEM if the relative fraction of Ce is low. Maybe an additional argument can be made based on the XPS data?

Response: Thank you for this comment. The related descriptions were clarified and explained, and the explanations have been revised in the manuscript:

If a CePt alloy is generated after laser irradiation, metallic Ce will appear. In general, in the XPS spectrum of Ce3d, the characteristic peaks located around 884.5 eV (Ce 3d_{3/2}) and 902.eV (Ce 3d_{5/2}) correspond to the metal Ce (Scientific Reports 2020, 10, 1. J. Phys. Chem. C 2007, 111, 3685). In our case, comparing the XPS spectra of Ce3d before and after laser irradiation, there were no characteristic peaks related to metallic Ce, which indicates that no CePt alloy phase was formed (Supplementary Fig. 5). To demonstrate it more closely, several large size Pt NPs with thick CeO_x overlays were selected for HRTEM analysis (Supplementary Fig. 6). No CePt alloy phase was observed at the interface between the Pt and CeO_x layers, which was in agreement with the XPS results.

Supplementary Fig 5. XPS spectra of Pt/CeO₂ and laser-irradiated Pt/CeO₂.

9) “According to the XPS results (Supplementary Fig. 9), the Pt/Ce ratio on the surface of specimens were 0.076 (Fresh-Pt/CeO₂), 0.0459 (Pt/CeO₂-laser-20 min), 0.0394 (Pt/CeO₂-laser-40 min), 0.0344 (Pt/CeO₂-laser-60 min), and 0.0557 (Pt/CeO₂-H₂), respectively. It means that Pt NPs were totally encapsulated with the increased exposure time, and the thickness of the overlayer was higher, indicating the enhanced structural reorganization” – define at what Pt/Ce ratio the encapsulation becomes “total”, and how this correlates with the argument that, according to the catalytic data, the overlayer is porous. In addition, I do not understand what “enhanced structural reorganization” means.

Response: Thank you for this comment. The related descriptions were clarified and explained, and the explanations have been revised in the manuscript:

(1) When increasing the exposure time, Pt NPs were more prone to be encapsulated. Indeed, it is not an appropriate expression about the word “totally”. In order to avoid confusion, the description of “According to the XPS results (Supplementary Fig. 9), the Pt/Ce ratio on the surface of specimens were 0.076 (Fresh-Pt/CeO₂), 0.0459 (Pt/CeO₂-laser-20 min), 0.0394 (Pt/CeO₂-laser-40 min), 0.0344 (Pt/CeO₂-laser-60 min), and 0.0557 (Pt/CeO₂-H₂), respectively. It means that Pt NPs were totally encapsulated with the increased exposure time, and the thickness of the

overlayer was higher, indicating the enhanced structural reorganization” was revised as “According to the XPS results (Supplementary Fig. 9), the Pt/Ce ratio on the surface of specimens were 0.076 (Fresh-Pt/CeO₂), 0.0459 (Pt/CeO₂-laser-20 min), 0.0394 (Pt/CeO₂-laser-40 min), 0.0344 (Pt/CeO₂-laser-60 min), and 0.0557 (Pt/CeO₂-H₂), respectively. It means that Pt NPs were more prone to be encapsulated with the increased exposure time, and the overlayers become thicker”. (Page 11)

(2) Due to the high-energy nature of the femtosecond laser, the overlayer induced with the laser is frequently discontinuous, and even a small number of Pt NPs are only partially encapsulated (Supplementary Fig. 7). The in-situ CO-DRIFT and H₂-TPD spectra demonstrate the porous nature of the CeO_x overlayer (Supplementary Fig. 24).

10) “After that, Pt NPs were loaded on the laser-treated CeO₂ and subjected to heat treatment.” – define the treatment.

Response: Thank you for your careful review of our manuscript. Your constructive comments and suggestions have helped us to further improve our manuscript: According to the comment, the description “After that, Pt NPs were loaded on the laser-treated CeO₂ and subjected to heat treatment” was revised as “After that, Pt NPs were loaded on the laser-treated CeO₂ and subjected to heat treatment (under argon atmosphere at 500°C for 2 h)”. (Page 12)

REVIEWERS' COMMENTS

Reviewer #2 (Remarks to the Author):

The Manuscript has further improved. I think that, overall, the chemical/catalytic part of the Manuscript is well-done and will be interesting to the community. I cannot judge if FDTD simulations have been done correctly or not -- this is beyond my expertise.

At the discretion of the Editor, this submission may be accepted for publication.